# DIFFERENTIALLY PRIVATE NETWORK TRAINING UNDER HIDDEN STATE ASSUMPTION

## ABSTRACT

We present a novel approach called differentially private stochastic block coordinate descent (DP-SBCD) for training neural networks with provable guarantees of differential privacy under the hidden state assumption. Our methodology regards neural networks as optimization problems and decomposes the training process of the neural network into sub-problems, each corresponding to the training of a specific layer. By doing so, we extend the analysis of differential privacy under the hidden state assumption to encompass non-convex problems and algorithms employing proximal gradient descent. Furthermore, in contrast to existing methods, we adopt a novel approach by utilizing calibrated noise sampled from adaptive distributions, yielding improved empirical trade-offs between utility and privacy.

## 1 INTRODUCTION

Machine learning models, especially deep neural networks, have exhibited remarkable progress in the last decade across diverse fields. Their applications, such as face recognition (Erkin et al., 2009) and large language models (Kandpal et al., 2022), have been integrated into people's daily lives. However, the increasing demand for large amounts of training data in the training process has given rise to growing concerns regarding the potential privacy vulnerabilities (Fredrikson et al., 2015; Shokri et al., 2017) associated with these models. For example, deep neural networks are shown to memorize the training data (Zhang et al., 2017) so that we can even reconstruct part of the training data from the learned model parameters (Haim et al., 2022). To address these issues, differential privacy (Dwork, 2006) has become the gold standard for making formal and quantitative guarantees on model's privacy and has been widely applied in learning problems (Abadi et al., 2016; Ha et al., 2019).

The predominate approach to ensuring differential privacy in the context of machine learning involves the incorporation of calibrated noise during each update step in the training phase, which leads to a trade-off between utility and privacy loss (Allouah et al., 2023; Alvim et al., 2012). Excessive noise often leads to a significant loss in utility, while insufficient noise may result in privacy leakage. Moreover, many privacy accountant methods, such as moment accountant (Abadi et al., 2016), are typically based on the composition properties of differential privacy. Specifically, they usually assume that the calibrated noise in each training iteration follows the same distribution and that the algorithm's internal states, i.e., the model's parameters in each training step, can be revealed to the adversaries. As a result, the total privacy loss of the algorithm significantly increases with the number of training iterations, because more internal states will be revealed with more training iterations. Such assumptions lead to a very loose privacy estimation of the algorithm, because, in practice, most internal states are usually not even recorded during training.

The gap between theory and practice motivates researchers to consider more practical assumptions. Recently, the hidden state assumption was proposed to narrow down this gap (Chourasia et al., 2021; Feldman et al., 2018; Ye and Shokri, 2022). This assumption posits that the internal states of the training phase are hidden, and that the adversaries only have access to the last iterate. Under this assumption, Chourasia et al. (2021); Ye and Shokri (2022) use Langevin diffusion to track the change rate of *Rényi Differential Privacy* (RDP) in each epoch and bound the privacy loss for *Differential Privacy Gradient Descent* (DP-GD) and *Differential Privacy Stochastic mini-batch Gradient Descent* (DP-SGD) (Abadi et al., 2016). Recently, Asoodeh and Diaz (2023) also considered directly using hockey-stick divergence instead of Rényi divergence. These works claim a converged and small

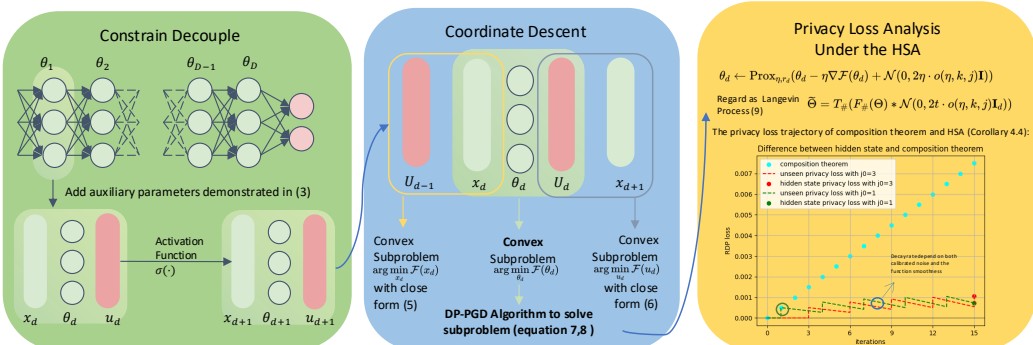

Figure 1: DP-SBCD framework. (Left) Use auxiliary parameters to decouple non-convex constraints and consider its Lagrangian function $\mathcal{F}(\boldsymbol{\theta}, \boldsymbol{x}, \boldsymbol{u})$.(Middle) sub-problems of DP-SBCD algorithm in solving the Lagrangian function.(Right) Privacy loss trajectory comparison for one sub-problem under the composition theorem and the hidden state assumptions.

privacy loss (usually less than 1) under some strict assumption such as strongly convex loss function and gradient Lipschitzness.

From a theoretical aspect, existing theorems (Feldman et al., 2018; Chourasia et al., 2021; Ye and Shokri, 2022) indicate a valuable property that privacy loss could *converge* under the hidden state assumption in learning problems with strongly convex and $\beta$-smooth loss functions. Otherwise, the privacy loss based on their analysis framework will *increase exponentially*. Consequently, these theorems cannot be directly applied to training deep neural networks, whose loss functions are non-convex and may exhibit non-smoothness when using activation functions like ReLU (Krizhevsky et al., 2012).

To extend existing analyses of differential privacy under the hidden state assumptions to deep learning problems, we propose *differentially private stochastic block coordinate descent* (DP-SBCD) to avoid the exponentially-increasing privacy loss for the traditional DP-SGD algorithm. By contrast, our analyses provide tight theoretical guarantees in privacy loss under the hidden state assumption when using DP-SBCD. As demonstrated in Figure 1, we first decouple the activation function to convert the task of training neural networks as a constrained optimization problem and subsequently construct the associated Lagrangian function. The DP-SBCD algorithm then considers each layer individually, decomposing the original learning problem into multiple sub-problems. The loss function for each of these sub-problems is convex, which facilitates our analyses to derive a tight privacy loss bound under the hidden state assumption. It is worth noting that our algorithm's privacy loss under the hidden state assumption is significantly smaller than that derived under the composition theory.

In addition, this work investigates the connections between the calibrated noise and the smoothness of the loss function in the context of differential privacy. Generally speaking, calibrated noise and the smoothness of the loss function both contribute to the privacy loss. However, how the calibrated noise influence the total privacy loss under the hidden state, which is still unknown so far, is valuable in improve the privacy-utility trade-off (Du et al., 2021). Intuitively speaking, we need noise of larger variance to achieve the same level of differential privacy if the Lipschitz constant of the loss gradient is bigger, but noise of large variance harms the utility when the algorithm approaches the optimal solution. This paper study the impact of the variance of the calibrated noise on differential privacy in different training phases and propose to adaptively adjust it during training. We also explain why the privacy loss will converge under the hidden state assumption. Empirically, our proposed algorithm with adaptive calibrated noise achieves a better trade-off between the model's utility and privacy.

In summary, we highlight our contributions as follows:

- We propose *differentially private stochastic block coordinate descent* (DP-SBCD), which, to the best of our knowledge, is the first feasible method to solve non-convex problems with differential privacy guarantees under the hidden state assumption.

- The privacy analysis in this work possess a generic nature, rendering them compatible with proximal gradient descent and adaptive calibrated noise. Our privacy analysis also answers why the privacy loss can converge (Ye and Shokri, 2022; Feldman et al., 2018) under the hidden state assumption.

- Our analysis and numerical experiments indicate that our algorithm's privacy loss under the hidden state assumption is minimal. Furthermore, by adaptively modifying the calibrated noise, our algorithm can achieve a better privacy-utility trade-off.

**Terminology and Notation** The privacy loss refers to the *Rényi differential privacy* (RDP). When using the diffusion process to analyze the update scheme, we use $t$ to represent the time in the process. We use $\| \cdot \|_2$ to represent $l_2$ norm for vectors and the spectral norm for matrices. In addition, $\| \cdot \|_F$ denotes Frobenius norm. Other notations are explicitly defined before usage.

## 2 RELATED WORKS

**Differential Privacy** Differential Privacy (DP) (Dwork, 2006) was initially introduced to quantitatively ensure the privacy guarantees within the realm of databases. Nevertheless, its rigorous mathematical foundations and nice properties, such as composition theorem and post-processing immunity, have raised more and more interest in its application in machine learning. Among the practical methods for achieving DP-guaranteed models, DP-SGD (Song et al., 2013) is the most popular one, which involves the addition of calibrated noise to the clipped per-sample gradients in each iterative update.

Besides the DP-guaranteed training algorithm, the proper accounting of privacy loss is also critical. Most accountant methods rely on the composition theorem of differential privacy. In this regard, Abadi et al. (2016) introduces the momentum accountant, Mironov (2017) further proposes Rényi differential privacy (RDP), which turns out a better tool to analyze the algorithm's privacy loss. Specifically, a randomized mechanism $\mathcal{M} : \mathcal{D} \to \mathcal{R}$ is said to have $(\alpha, \varepsilon)$-RDP if for any adjacent datasets $d, d' \in \mathcal{D}$, it holds that $D_\alpha(\mathcal{M}(d)||\mathcal{M}(d')) \leq \varepsilon$ where $D_\alpha$ denotes the Rényi divergence of order $\alpha$.

Recently, the hidden state assumption has been explored in many works (Chourasia et al., 2021; Feldman et al., 2018; Ye and Shokri, 2022). While the composition theory assumes that adversaries have access to the intermediate states of the algorithm, which requires the summation of the privacy loss across each training iteration (Ponomareva et al., 2023), the hidden state assumption posits that these intermediate states remain concealed. In real-life applications, the intermediate states usually cannot be tracked, so the hidden state assumption is more practical. It is first proposed in (Feldman et al., 2018), which also proves that a sequence of contractive maps can exhibit bounded $(\alpha, \varepsilon)$-RDP. Subsequently, Chourasia et al. (2021) consider the update scheme as a Langevin diffusion process and analyze the privacy loss for DP-GD using the log-Sobolev inequality (Vempala and Wibisono, 2019). Expanding upon this, Ye and Shokri (2022) analyze DP-SGD by leveraging the convergent analysis of unadjusted Langevin algorithm proposed by Wibisono (2019).

**Block Coordinate Descent** The block coordinate descent algorithm is designed to decompose the original problem into several sub-problems which allow for independent and efficient solutions. It not only helps prevent the vanishing gradient problem (Zhang and Brand, 2017) but also facilitates distributed or parallel implementations (Mahajan et al., 2017). In the era of deep learning, many works focus on employing block coordinate descent to train neural networks (Gu et al., 2020; Zeng et al., 2019; Zhang and Brand, 2017). Zhang and Brand (2017) utilize a lifting trick to solve the learning problem of deep neural networks with ReLU activation function, which is then extended by Gu et al. (2020); Mangold et al. (2022). Furthermore, Zeng et al. (2019) comprehensively analyzes two general types of optimization formulas: Two-splitting and Three-splitting. They also investigate the loss function, activation function, and architecture that these algorithms require.

## 3 SOLVING NEURAL NETWORK USING BLOCK COORDINATE DESCENT

In this section, we regard the deep neural network as a optimization problem and use auxiliary parameters to decouple the input and the output of the activation function. Then, we use the block coordinate descent and analyze the properties of the decomposed sub-problems.

## 3.1 BLOCK COORDINATE DESCENT ALGORITHM

We consider learning an $D$-layer neural network parameterized by $\boldsymbol{\theta} \overset{\text{def}}{=} \{\theta_d\}_{d=0}^{D}$:

$$\min_{\boldsymbol{\theta}} \mathcal{L}(\boldsymbol{\theta}, \boldsymbol{x}) \overset{\text{def}}{=} \mathcal{R}\left(\theta_D \sigma_{D-1}\left(\theta_{D-1}(\ldots \sigma_1(\theta_0 \boldsymbol{x}_0))\right); y\right) \tag{1}$$

Here, $\boldsymbol{x} \overset{\text{def}}{=} \boldsymbol{x}_0$ is the input, $\mathcal{R}$ is the loss function, such as softmax cross-entropy function, which is convex w.r.t. $\theta_D$. Moreover, $\{\sigma_d\}_{d=0}^{D-1}$ are activation functions, such as ReLU function, that are Lipschitz continuous. Finally, $\{\theta_d\}_{d=0}^{D}$ are matrices representing the weights of linear layers, including fully-connected layers and convolutional layers.

Note that the function defined in Equation (1) is highly non-convex as the variables are coupled via the deep neural networks architecture, which is challenging in privacy loss analysis under the hidden state assumption. Therefore, we first add auxiliary variables and convert Problem (1) as a constrainted optimization problem(Zeng et al., 2019):

$$\min_{\boldsymbol{\theta}} \mathcal{L}(\boldsymbol{\theta}, \boldsymbol{x}) := \mathcal{R}(\theta_D \boldsymbol{x}_D; y) \quad \text{s.t.} \quad \boldsymbol{x}_{d+1} = \sigma_d(\boldsymbol{u}_d), \ \boldsymbol{u}_d = \theta_d \boldsymbol{x}_d \quad d = 0, \ldots, D-1 \tag{2}$$

where $\{\boldsymbol{x}_d\}_{d=1}^{D}$ and $\{\boldsymbol{u}_d\}_{d=0}^{D-1}$ are vectors representing the intermediate activations of each layer. Problem (2) introduces auxiliary variables and constraints to decouple variables in Problem (1). Then, we consider the Lagrangian function of Problem (2) with a multiplier co-efficient $\gamma$, which is set 1 in this work.

$$\mathcal{F}(\boldsymbol{\theta}, \boldsymbol{x}, \boldsymbol{u}) = \mathcal{R}(\theta_D \boldsymbol{x}_D; y) + \frac{\gamma}{2} \sum_{d=0}^{D-1} (\|\boldsymbol{x}_{d+1} - \sigma_d(\boldsymbol{u}_d)\|_2^2 + \|\boldsymbol{u}_d - \theta_d \boldsymbol{x}_d\|_2^2) \tag{3}$$

where $\mathcal{F}$ is a function of $\{\theta_d\}_{d=0}^{D}$, $\{\boldsymbol{x}_d\}_{d=0}^{D}$ and $\{\boldsymbol{u}_d\}_{d=0}^{D-1}$. For notation simplicity, we only explicitly highlight the parameter we consider for $\mathcal{F}$ if there is no ambiguity. For example $\mathcal{F}(\theta_d')$ means we update the parameter $\theta_d$ in $\mathcal{F}$ while keeping the other parameters fixed. We use block coordinate descent to update parameters of $\mathcal{F}$ where we treat each layer as a sub-problem. The algorithm is summarized in Algorithm 1. Considering the loss function $\mathcal{R}$ is a convex function w.r.t. both $\theta_D$ and $\boldsymbol{x}_D$, we have for all $d$, $\mathcal{F}(\theta_d)$ is convex w.r.t. $\theta_d$, and $\forall d$, $\mathcal{F}(\boldsymbol{x}_d)$ is convex w.r.t. $\boldsymbol{x}_d$.

To update $\theta_d$, Algorithm 1 also considers the damping term $\frac{1}{2\eta}\|\theta_d - \theta_d'\|_F^2$ and the regularization term $r_d(\theta_d')$, which can be optimized by the proximal gradient descent elaborated in the next section. To update $\boldsymbol{x}_d$, we have the analytical solution due to the convexity of $\mathcal{F}(\boldsymbol{x}_d)$.

$$\boldsymbol{x}_d \leftarrow \begin{cases} \left(\theta_d^T \theta_d + \mathbf{I}\right)^{-1} \left(\theta_d^T \boldsymbol{u}_d + \sigma(\boldsymbol{u}_{d-1})\right) & \text{if } d = 0, 1, ..., D-1. \\ \text{Prox}_{\frac{1}{\gamma}, \mathcal{R}}(\sigma_{d-1}(\boldsymbol{u}_{d-1})). & \text{if } d = D \end{cases} \tag{4}$$

Additional, we clip the auxiliary $\boldsymbol{x}_d$ by the threshold $\rho_d$, we will further explain it in the next section.

$$\boldsymbol{x}_d \leftarrow \boldsymbol{x}_d \cdot \min(\rho_d/\|\boldsymbol{x}_d\|_2, 1) \tag{5}$$

To update $\boldsymbol{u}_d$, we utilize Zeng et al. (2019, Lemma 13) as follows to obtain the analytical solution.

$$\boldsymbol{u}_d \leftarrow \begin{cases} \theta_d \boldsymbol{x}_d & \text{if} - \boldsymbol{x}_{d+1} \leq \theta_d \boldsymbol{x}_d \leq -(\sqrt{2}-1)\boldsymbol{x}_{d+1} \leq 0. \\ \min\{0, \theta_d \boldsymbol{x}_d\} & \text{if } \theta_d \boldsymbol{x}_d + \boldsymbol{x}_{d+1} \leq 0. \\ \frac{1}{2}\left(\theta_d \boldsymbol{x}_d + \boldsymbol{x}_{d+1}\right) & \text{otherwise.} \end{cases} \tag{6}$$

## 3.2 THE SMOOTHNESS OF THE SUB-PROBLEMS

In this subsection, we analyze the smoothness of the sub-problem $\mathcal{F}(\theta_d)$. In our algorithm, by clipping $\boldsymbol{x}_d$, we bound each layer's output $x_d$.Then, we have the following lemma shows that the sub-problem $\mathcal{F}(\theta_d)$ as a function of $\theta_d$ is $\beta_d$-smooth with the exact sommthness constant.

**Algorithm 1** Stochastic Block Coordinate Descent For Neural Networks

**Input:** training set, regularization schemes $\{r_d\}_{d=0}^{D}$, parameter $\eta$, batch size $b$, clipping threshold $\rho_d$.
**Initialization:** $\{\theta_d\}_{d=0}^{D}, \{\boldsymbol{x}_d\}_{d=0}^{D}, \{\boldsymbol{U}_d\}_{d=0}^{D-1}$.
**for** epoch $k = 0, 1, \ldots, K-1$ **do**
  **for** each mini-batch of size $b$ **do**
    **for** layer $d = 0, 1, \ldots, D$ **do**
      Update $\theta_d, \boldsymbol{u}_d, \boldsymbol{x}_d$ simultaneously as follows:
      $\boldsymbol{x}_d \leftarrow \arg\min_{\boldsymbol{x}'_d} \mathcal{F}(\boldsymbol{x}'_d)$
      Clip $\boldsymbol{x}_d$ by (5) with coefficient $\rho_d$.
      $\boldsymbol{u}_d \leftarrow \arg\min_{\boldsymbol{u}'_d} \mathcal{F}(\boldsymbol{u}'_d)$
      $\theta_d \leftarrow \arg\min_{\theta'_d} \mathcal{F}(\theta'_d) + \frac{1}{2\eta}\|\theta_d - \theta'_d\|_F^2 + r_d(\theta'_d)$
    **end for**
  **end for**
**end for**

**Algorithm 2** Differentially Private Stochastic Block Coordinate Descent For Neural Networks

**Input:** training set, regularization schemes $\{r_d\}_{d=0}^{D}$, step size $\eta$, batch size $b$, noise control function $o(\eta, k, j)$, clipping threshold $\rho_d$.
**Initialization:** $\{\theta_d\}_{d=0}^{D}, \{\boldsymbol{x}_d\}_{d=0}^{D}, \{\boldsymbol{U}_d\}_{d=0}^{D-1}$.
**for** epoch $k = 0, 1, \ldots, K-1$ **do**
  **for** each mini-batch of size $b$ **do**
    **for** layer $d = 0, 1, \ldots, D$ **do**
      Update $\theta_d, \boldsymbol{u}_d, \boldsymbol{x}_d$ simultaneously as follows:
      $\boldsymbol{x}_d \leftarrow \arg\min_{\boldsymbol{x}'_d} \mathcal{F}(\boldsymbol{x}'_d)$
      Clip $\boldsymbol{x}_d$ by (5) with coefficient $\rho_d$.
      $\boldsymbol{u}_d \leftarrow \arg\min_{\boldsymbol{u}'_d} \mathcal{F}(\boldsymbol{u}'_d)$
      Update $\theta_d$ based on (8)
    **end for**
  **end for**
**end for**

**Lemma 3.1.** *If the input $\boldsymbol{x}_d$ is bounded with $\rho_d$ and the activation function is ReLU, then the function $\mathcal{F}(\theta_d)$ for any layer $0 \leq d \leq D$ is $\beta_d$-smooth and the smoothness constant $\beta_d$ is $\gamma\rho_d^2$.*

The proof is deferred to Appendix A.1. Lemma 3.1 shows that for Algorithm 2, we could control the smoothness sub-problem $\mathcal{F}(\theta_d)$ by $\rho_d$. Moreover, We have to highlight that the $\mathcal{F}(\theta_d)$ **assumes that other parameters except $\theta_d$ are fixed**. In addition, Lemma 3.1 indicates that the smoothness constant $\beta_d$ does not depend on other parameters. Based on this, we can calculate the Hessian matrix of $\mathcal{F}(\theta_d)$ as $\nabla^2 \mathcal{F}(\theta_d) = \boldsymbol{x}_d\boldsymbol{x}_d^T \geq 0$. Therefore, $\mathcal{F}(\theta_d)$ is convex w.r.t. $\theta_d$.

# 4 DIFFERENTIALLY PRIVATE STOCHASTIC BLOCK COORDINATE DESCENT

In this section, we propose *Differentially Private Stochastic Block Coordinate Descent* (DP-SBCD), i.e., the differentially private version of Algorithm 1. We then calculate the algorithm's privacy loss under the hidden state assumption. We discuss the privacy loss in a generic form, especially the case when using adaptive calibrated noise.

## 4.1 DIFFERENTIALLY PRIVATE UPDATE SCHEME

We first consider the update scheme for $\theta_d$ in Algorithm 1 and use first order Taylor expansion of $\mathcal{F}(\theta'_d)$ to write it in the proximal gradient descent format.

$$\theta_d \leftarrow \underset{\theta'_d}{\arg\min} \langle \nabla\mathcal{F}(\theta_d), \theta'_d - \theta_d \rangle + \frac{1}{2\eta}\|\theta'_d - \theta_d\|_F^2 + r_d(\theta'_d) = \text{Prox}_{\eta, r_d}\left(\theta_d - \eta\nabla\mathcal{F}(\theta_d)\right) \quad (7)$$

Owing to the post-processing immunity of differential privacy, the privacy guarantee is required for each sub-problem. Furthermore, as the algorithm converges gradually, the gradient $\nabla\mathcal{F}(\theta_d)$ diminishes, prompting the use of calibrated noise with an adaptive variance instead of a fixed one. As a result, we propose the following update scheme:

$$\theta_d \leftarrow \text{Prox}_{\eta, r_d}\left(\theta_d - \eta\nabla\mathcal{F}(\theta_d) + \mathcal{N}(0, 2\eta \cdot o(\eta, k, j)\,\mathbf{I})\right) \quad (8)$$

where $\mathcal{N}(0, \mathbf{I})$ is the standard Gaussian distribution, $o(\eta, k, j)$ is a function of the step size $\eta$, epoch index $k$ and iteration index $j$ to control the magnitude of the calibrated noise. By incorporating the update scheme (8) into Algorithm 1, we obtain Algorithm 2, which will be proved to guarantee differential privacy in the following section.

## 4.2 PRIVACY LOSS OF SUB-PROBLEMS

We now study the privacy loss of Algorithm 2. Leveraging the post-processing immunity and composition property, we can establish an upper bound of the privacy loss of Algorithm 2 by

summing the privacy losses of its individual sub-problems. Consequently, our primary focus here is to estimate the privacy loss associated with each sub-problem. For notation simplicity, we omit the subscript $d$ in this subsection, as our analyses apply to any sub-problem.

In each iteration of Algorithm 2, the value of $\theta$ is updated by (8) and then normalized by (5). Among them, it is clear that the scaling operation of $\theta$ in (5) does not contribute to the privacy loss, so we focus on the update scheme (8).

We regard the update scheme (8) in Algorithm 2 as a diffusion process (Balle et al., 2019; Chourasia et al., 2021; Ye and Shokri, 2022). Specifically, the update scheme consists of three parts: The gradient descent part $\theta - \eta\nabla\mathcal{F}(\theta)$; The noise part $o(\eta, k, j)$; The proximal operator associated with a convex regularization function $r$. From a distributional perspective, let $\Theta$ be the distribution of the parameter $\theta$, then the distribution of the parameter after one-step update in (8) can be represented as follows:

$$\widetilde{\Theta} = T_{\#}(F_{\#}(\Theta) * \mathcal{N}(0, 2t \cdot o(\eta, k, j)\mathbb{I}_d)) \tag{9}$$

where $F_{\#}$ and $T_{\#}$ are two push-forward mappings. $F$ represents the gradient descent update, $T$ represents the proximal operator, and $*$ represents the convolution operator between two distributions.

Based on the smoothness property of $\mathcal{F}(\theta)$ indicated in Lemma 3.1, we prove the Lipschitz continuity of the first part of the update scheme (8).

**Lemma 4.1** (Lipschitz continuity for $F$). *If $\mathcal{F}(\theta)$ is a $\beta$-smooth function and $\nabla^2\mathcal{F}(\theta) \geq \omega$, then the update function $F(\theta) = \theta - \eta\nabla\mathcal{F}(\theta)$ is Lipschitz continuous with a constant $L_F \leq \max\{|1 - \eta\omega|, |1 - \eta\beta|\}$.*

The proof is deferred to Appendix A.2. We also need to highlight that the $\mathcal{F}(\theta)$ in the Lemma 4.1 **assumes that other parameters except $\theta$ are fixed**. In addition, the bound of the Lipschitz constant derived in Lemma 4.1 does not depend on other parameters. That is to say, the bound in Lemma 4.1 is valid for arbitrary values of other parameters. Since $\mathcal{F}(\theta)$ is proven convex in Section 3, $0 \leq \omega \leq \beta$. As a result, the Lipschitz constant will be dominated by the convexity term $|1 - \eta\omega|$ when the step size $\eta \leq \frac{2}{\omega+\beta}$ and otherwise the smoothness term $|1 - \eta\beta|$.

Similarly, we can prove the Lipschitz continuity of the third part of the update scheme (8).

**Lemma 4.2** (Lipschitz continuity for $T$). *If $\eta > 0$ and $r(\theta)$ is a convex function, then the proximal operator function $T(\theta) = Prox_{\eta,r}(\theta) = \arg\min_{\widetilde{\theta}}\left\{r(\widetilde{\theta}) + \frac{1}{2\eta}\|\widetilde{\theta} - \theta\|_2^2\right\}$ is Lipschitz continuous with a constant $L_T \leq 2$.*

The proof is deferred to Appendix A.3. Typical examples of the regularization function $r$ include 1) no regularization: $r(\theta) = 0$, and then $Prox_{\eta,r}(\theta) = \theta$; 2) $l_2$ regularization in weight decay: $r(\theta) = \frac{1}{2}\|\theta\|_2^2$, and then $Prox_{\eta,r}(\theta) = \frac{\eta}{1+\eta}\theta$; 3) $l_1$ regularization in LASSO: $r(\theta) = \|\theta\|_1$, and then $Prox_{\eta,r}(\theta) = sign(\theta) \cdot \max(0, |\theta| - \eta)$. In all these three cases, the proximal function is Lipschitz continuous and the Lipschitz constant is 1.

Now we assume that $\theta$ for each sub-problem satisfies the log-Sobolev inequality (LSI), which is a benign assumption used in Vempala and Wibisono (2019); Ye and Shokri (2022) for the distribution of parameter $\theta$. The formal definition of log-Sobolev inequality is provided in Definition A.1 in the appendix for reference. The LSI assumption is very mild, Vempala and Wibisono (2019) shows that strongly log-concave distribution, such as Gaussian distribution, uniform distribution, and some non-logconcave distribution satisfy the LSI assumption. Based on the log-Sobolev inequality assumptions and Lipschitzness (Lemma 4.1,4.2), we consider the recursive privacy dynamics for Equation (9) and bound the change rate of RDP during one step of noisy mini-batch proximal gradient descent. Our formal result is demonstrated as Theorem A.2 in Appendix A.4. It is an extension of Ye and Shokri (2022, Lemma 3.2) to the cases of non-convex loss functions, proximal gradient descent and adaptive calibrated noise.

To derive the privacy loss of Algorithm 2, we assume a bounded sensitivity of the total gradient for sub-problem $\mathcal{F}(\theta)$, which is popular used in (Ye and Shokri, 2022; Das et al., 2023).

**Assumption 4.3.** *(Sensitivity of the Total Gradient) The $l_2$ sensitivity of the total gradient $\mathbb{E}_D\nabla\mathcal{F}(\theta)$ is finite. That is to say, $\exists S_g < +\infty$ such that for any dataset $D$ and its neighbouring dataset $D'$ that only differs in one instance, we have $S_g = \max_{D,D',\theta}\|\mathbb{E}_D\nabla\mathcal{F}(\theta) - \mathbb{E}_{D'}\nabla\mathcal{F}(\theta)\|_2$.*

Based on Lemma 3.1, $S_g = \gamma\rho$. Then, we apply the Theorem A.2 to the case of two neighboring datasets and directly obtain the privacy loss of the Algorithm 2 for each iteration. The proof is nearly the same as Ye and Shokri (2022, Lemma 3.2), we omit the proof detail for the sake of brevity.

**Corollary 4.4.** *Under Assumption 4.3 where the sensitivity of the total gradient is $S_g < +\infty$, let $D$, $D'$ be an arbitrary pair of the neighboring datasets that only differ in the $i_0$-th data point (i.e. $x_{i_0} \neq x'_{i_0}$). Let $B_k^j$ be a fixed mini-batch used in the $j$-th iteration of the $k$-th epoch in Algorithm 2, which contains $b$ different training instances whose indices are sampled from $\{0, 1, \ldots, n-1\}$. We denote $\theta_k^j$ and $\theta'^{\,j}_k$ as the intermediate parameters in Algorithm 2 when using datasets $D$ and $D'$, respectively. If the distributions of $\theta_k^j$ and $\theta'^{\,j}_k$ satisfy log-Sobolev inequality with a constant $c$, the update function $F(\theta) = \theta - \eta\nabla\mathcal{F}(\theta)$ and the proximal operator $T(\theta) = Prox_{\eta,r}(\theta)$ are Lipschitz continuous with Lipschitz constant $L_F$ and $L_T$, respectively, then the following recursive bound for Rényi divergence holds for any order $\alpha > 1$:*

*1) If $i_0 \notin B_k^j$, then $\frac{R_\alpha(\theta_k^{j+1}||\theta'^{\,j+1}_k)}{\alpha}$ is upper bounded by $\frac{R_{\alpha'}(\theta_k^j||\theta'^{\,j}_k)}{\alpha'} \cdot \left(1 + \frac{c \cdot 2\eta \cdot o(\eta,k,j)}{L_F^2}\right)^{-1/L_T^2}$ where the order $\alpha' = (\alpha - 1)\left(1 + \frac{c \cdot 2\eta o(\eta,k,j)}{L_F^2}\right)^{-1} + 1$.*

*2) If $i_0 \in B_k^j$, then $\frac{R_\alpha(\theta_k^{j+1}||\theta'^{\,j+1}_k)}{\alpha}$ is upper bounded by $\frac{R_\alpha(\theta_k^j||\theta'^{\,j}_k)}{\alpha} + \frac{\eta S_g^2}{4b^2 \cdot o(\eta,k,j)}$.*

Compared with the results discussed in Ye and Shokri (2022) which only study the case without proximal operator, the privacy loss decay rate in the first case of Corollary 4.4 is powered by $-1/L_T^2$ instead of $-1$, corresponding to the factor $1/L_T^2$ in the formulation of $c_t$ in the change rate of RDP in Theorem A.2. This indicates that the Lipschitz constant $L_T$ of the proximal operator also affects the privacy loss decay, a.k.a. privacy amplification when we run Algorithm 2 in the hidden state assumption. The smaller $L_T^2$ is, the better privacy guarantee will be obtained. Corollary 4.4 concludes the change of the privacy loss for one iteration in Algorithm 2.

By applying Corollary 4.4 iteratively, we can obtain the algorithm's privacy loss for the whole training phase. The formal theorem is demonstrated below.

**Theorem 4.5.** *Under Assumption 4.3 where the sensitivity of the total gradient is $S_g < +\infty$, the distribution of $\theta$ satisfies log-Sobolev inequality with a constant $c$. In addition, the update function $F(\theta) = \theta - \eta\nabla\mathcal{F}(\theta)$ and the proximal operator $T(\theta) = \mathrm{Prox}_{\eta,r}(\theta)$ are Lipschitz continuous with Lipschitz constant $L_F$ and $L_T$, respectively. If we use Algorithm 2 to train model parameters $\theta$ for $K \geq 1$ epochs, then the algorithm satisfies $(\alpha, \varepsilon(\alpha))$-Rényi differential privacy with the constant:*

$$\varepsilon_K(\alpha) \leq \frac{1}{\alpha - 1} \log\left(\sum_{j_0=0}^{n/b-1} \frac{b}{n} \cdot e^{(\alpha-1)(\varepsilon_K(\alpha, j_0))}\right)$$

*where:*

$$\varepsilon_K(\alpha, j_0) \leq \alpha \sum_{k=0}^{K-1} \frac{\eta S_g^2}{4b^2 \cdot o(\eta, k, j_0)} \cdot \left(\frac{c_k^{j_0+1}}{c_K^{n/b-1}}\left(\frac{1}{L_F^2 L_T^2}\right)^{(n/b-1)(K-k)-j_0} \prod_{l=k}^{K} \frac{c_l^{j_0+1}}{c_l^{j_0}}\right)^{-1/L_T^2} \tag{10}$$

*In Inequality (10), $c_k^j$ is the log-Sobolev inequality constant for the distribution of the model parameters in the $j$-th iteration of the $k$-th epoch. The value of $c_k^j$ is calculated based on Lemma A.3 in the appendix.*

*Proof Sketch.* For each iteration of the Algorithm 2, the update scheme (8) fixed parameters except the $\theta$. Hence, although the $\mathcal{F}(\theta)$ in the update scheme differs among each iteration because of different fixed parameters, it maintains Lipschitzness in each iteration. Hence, we could use Corollary 4.4 to analyze the algorithm's privacy loss. In each epoch, there is one and only one different mini-batch for two neighboring datasets, so we assume it is the $j_0$-th batch w.l.o.g. and apply case 2) of Corollary 4.4 for this mini-batch update and case 1) of Corollary 4.4 for the other updates. For any $\alpha$, the original privacy loss is 0 and the recursive bound in Corollary 4.4 holds. Therefore, we can uniformly bounded the value of $\frac{\varepsilon_K(\alpha, j_0)}{\alpha}$ for any $\alpha$. Finally, since $j_0$ is uniformly distributed among the index set $\{0, 1, \ldots, n/b - 1\}$, we use the joint convexity of scaled exponentiation of Rényi divergence to bound the final privacy loss $\varepsilon_K(\alpha)$. The complete proof is deferred to the AppendixA.5. $\square$

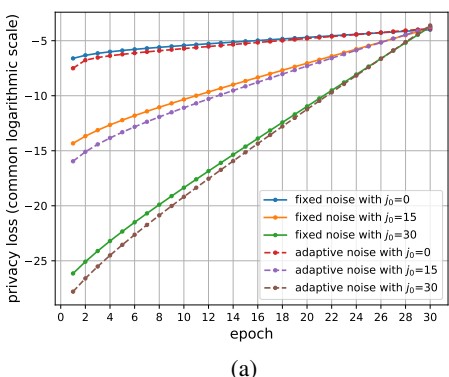 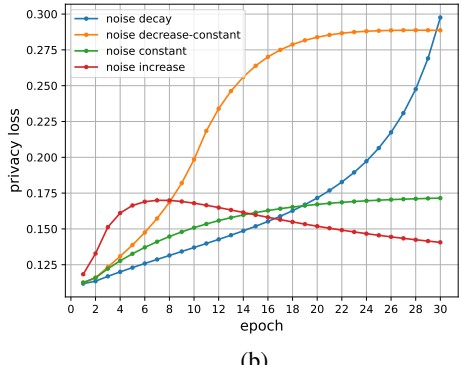

(a)  (b)

Figure 2: Common hyper-parameter settings: batch size $b = 100$, stepsize $\eta = 0.01$, $x_d$ threshold $\rho = 1$, data set $n = 2100$, RDP order $\alpha = 10$. (a) Each epoch's privacy loss contribution for the Algorithm 2 for $K = 30$ epochs. We study the scenarios of both fixed and adaptive noise. Each scenario contains three situations($j_0 = 0, 15, 30$). In the fixed noise scenario, the $o(\eta, k, j) = 0.01$, while in the adaptive noise scenario, the $o(\eta, k, j) = 0.01 - 0.003k$. (b) Privacy loss for different numbers of epochs $K$ for Algorithm 2. We show four scenarios: noise decay means $o(\eta, k, j) = 0.001 - 0.0003k$, noise constant means $o(\eta, k, j) = 0.0005$, noise increase means $o(\eta, k, j) = 0.0001 + 0.0003k$, and noise decrease-constant means that $o(\eta, k, j) = 0.0005 - 0.00003k$ where $k \in [1, 10]$ and remain $o(\eta, k, j) = 0.0002$ where $k \in [10, 30]$.

$\varepsilon_K(\alpha, j_0)$ in the Theorem 4.5 represents the privacy loss when the only different instance of the two neighboring datasets is in the $j_0$-th mini-batch of each epoch. The $\varepsilon_K(\alpha, j_0)$ shows that the overall privacy loss is the summation of each epoch's privacy loss term $\frac{\eta S_g^2}{4b^2 \cdot o(\eta, k, j)}$ times a decay rate term $\left( \frac{c_k^{j_0+1}}{c_K^{n/b-1}} \left( \frac{1}{L_F^2 L_T^2} \right)^{(n/b-1)(K-k)-j_0} \prod_{l=k}^{K} \frac{c_l^{j_0+1}}{c_l^{j_0}} \right)^{-1/L_T^2}$. More importantly, Algorithm 2 maintains the decay rate term smaller than 1 and decreases with the increase in $K$.

Theorem 4.5 improves the results in existing works from many aspects for estimating the differential privacy under hidden state assumptions. Firstly, it is more generally applicable to different algorithms. Theorem 4.5 could easily extend the analyses and privacy guarantees from the gradient descent algorithm with calibrated noise from a fixed distribution to proximal gradient descent with adaptive calibrated noise. Second, it provides a feasible privacy loss accountant for non-convex problems with adaptive noise. In contrast to the assumption of strong convexity and $\beta$-smoothness on the objective function in existing results, Theorem 4.5 shows that Algorithm 2 applicable to general neural networks with Lipschitz constraints. Finally, even when downgrading to the case of gradient descent with calibrated noise sampled from fixed distributions, Theorem 4.5 has a tighter bound than previous work (Ye and Shokri, 2022, Theorem 3.3). This is because we directly bound $\varepsilon_K(\alpha, j_0)$ by recursively applying Corollary 4.4. By contrast, Ye and Shokri (2022) approximate the bound $\varepsilon_K(\alpha, j_0)$ by part of iterations in one epoch rather than the whole epoch. Corollary 4.4 shows an exponential decrease in privacy loss, so such approximation affects the final privacy loss accountant, we draw a Figure in the Appendix to indicate the difference.

### 4.3 PRIVACY LOSS VARIATION UNDER THE ADAPTIVE NOISE

As discussed above, the bound of $\varepsilon_K(\alpha, j_0)$ in Theorem 4.5 elucidates how each epoch influences the cumulative privacy loss, exhibiting a decay rate. Specifically, these contributions are inversely proportional to $o(\eta, j, k)$, i.e., the variance of the noise, and decrease exponentially [1] as the number of iterations and epochs increase. That is to say, under the hidden state assumption, the calibrated noise in the last few epochs primarily contributes to the total privacy loss.

---

[1]While not strictly exponential due to variations in the log-Sobolev inequality constant, the behavior closely resembles that of an exponential function curve in simulations.

We illustrate this phenomenon in Figure 2(a), which encompasses scenarios with both constant and non-constant $o(\eta, j, k)$ values. The numerical results align with the analysis, as indicated by the near-linear curves observed in the logarithmic scale graph. Furthermore, we observe that a larger value of $j_0$, signifying a delayed occurrence of the mini-batch containing the unique instance, leads to a smaller privacy loss during the initial stages of training. However, as the training progresses, this disparity diminishes. This observation is consistent with Theorem 4.5, as the privacy loss is zero for the first $j_0$ mini-batches of the first epoch.

Theorem 4.5 also enhances our understanding of the *convergence of the privacy loss*, initially proposed in Feldman et al. (2018): when $o(\eta, k, j)$ is a constant, an equilibrium solution emerges between the privacy loss and the decay rate after several epochs, resulting in the convergence of privacy loss under the hidden state assumption. In the case of adaptive calibrated noise, Figure 2(b) shows the tendency of privacy loss under four distinct noise settings. Empirical findings suggest that privacy loss converges when a constant magnitude of noise is utilized in the later stages of training. Conversely, the privacy loss diverges if the noise magnitude continues to decrease in the late phase of training. In contrast, increasing the noise magnitude can even lead to a decrease in privacy loss. However, it is worth noting that noise with a smaller variance tends to yield better utility for the model compared to noise with a larger variance, which is consistent with the utility-privacy trade-off (Allouah et al., 2023; Alvim et al., 2012).

## 5 DISCUSSION AND FUTURE WORKS

Feldman et al. (2018); Ye and Shokri (2022) shows that the privacy loss can converge under the hidden state assumption for strongly convex problem. However, such appealing result cannot be applied to neural network training because of the high non-convexity of the corresponding loss function (1). In this work, instead of directly use the gradient descent in training, we proposed the *Differentially Private Stochastic Block Coordinate Descent* (DP-SBCD) algorithm and calculate the corresponding privacy loss under the hidden state assumption, which is tight and generally applicable. To the best of our knowledge, our method is the first feasible method to solve non-convex problems with differential privacy guarantees under the hidden state assumption. We also implement our algorithm on various of datasets in Appendix B.1 to show the validation of our algorithm.

Furthermore, our privacy loss analysis further explains why the privacy loss will converge under the hidden state assumption. As demonstrated in the Section 4.3, the privacy loss under the hidden state assumption does not actually converge. An equilibrium solution emerge once the calibrated noise is fixed, and it will modify once the calibrated noise is changed. Moreover, Inspired by the fact Yu et al. (2019); Du et al. (2021) that adaptively allocate privacy budget could improve the utility-privacy trade-off. Our algorithm also consider adaptively modify the calibrated noise. Although our paper does not provide further theoretical analysis on what is the optimal adaptive calibrated noise due to the page limitation, the experiment in Appendix B.2 shows that adaptively choosing the calibrated noise could empirically provide a better utility-privacy trade-off under the hidden state assumption.

Although our algorithm has a lower privacy loss than DP-SGD, it's important to note that direct comparisons of privacy loss may be misleading because of different privacy loss assumptions: DP-SBCD is under the hidden state assumption and the DP-SGD is under the composition theorem. As demonstrated in the Section 4.3, the privacy loss under the hidden state assumption is predominated by the last few epochs, but the privacy loss under the composition theorem is proportional to the training iterations. In the Appendix B.1, we list the privacy loss of DP-SGD to provide an intuitive comparison. Moreover, the DP-SBCD algorithm and the DP-SGD algorithm have totally different parameter settings, such as the optimal step size and batch size, which is important in both the privacy loss accountant and the algorithm's utility.

Compare with the SGD algorithm, the coordinate descent algorithm requires less memory in training process because it only update one *block* of the model parameters. Recently, Luo et al. (2024) shows that the coordinate descent algorithm can reduce the memory consumption in large language training. However, coordinate descent also introduces new challenges: the stochastic of mini-batch and the block-wise parameter update leads to a high variance of the mini-batch gradient. In our experiment, we use large batch size to effectively mitigate the high variance of the mini-batch gradient, which increases memory consumption and might influence the utility. Although there are many works discussing variance reduction (Ding and Li, 2020; Gorbunov et al., 2020; Ding and Li, 2021) and large

batch training technique (Keskar et al., 2016; Hoffer et al., 2017), further improving our algorithm requires rigorously analyzing the algorithm's convergence and exploring the integration of these techniques. We leave this as future works.

## 6 CONCLUSION

We propose *differentially private stochastic block coordinate descent* (DP-SBCD) algorithm, which includes proximal gradient descent and adaptive noise, to train neural networks. As far as we are aware, DP-SBCD is the first algorithm capable of addressing non-convex training problems while ensuring a tight differential privacy guarantee under the hidden state assumption. Our theoretical analyses indicate different contributions of privacy loss in different training phases under the hidden state assumption, inspiring the idea of adaptively adjusting the calibrated noise during training. The adaptive noise proposed in our method can provide adjustable trade-offs between the model's utility and privacy. Moreover, our observations indicate that under proper settings, DP-SBCD could provide a better trade-off between utility and privacy. Going forward, our future research will concentrate on further refining the convergence and performance of the algorithm.

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

## A  PROOFS

### A.1  PROOF OF LEMMA 3.1

*Proof.* We first calculate the gradient of $\mathcal{F}(\theta_d)$ as follows:

$$\nabla\mathcal{F}(\theta_d) = \gamma(\boldsymbol{U}_d - (\theta_d\boldsymbol{x}_d))\boldsymbol{x}_d^T$$

where $\boldsymbol{x}_d$ and $\boldsymbol{U}_d$ is fixed. Therefore, $\forall\theta_d, \theta_d'$, we have the following bound:

$$\|\nabla\mathcal{F}(\theta_d) - \nabla\mathcal{F}(\theta_d')\|_2 \leq \gamma\|\boldsymbol{x}_d\|_2^2\|\theta_d - \theta_d'\|_2 \tag{11}$$

Let $\|\boldsymbol{x}_d\|_2 \leq X_d$, so the smoothness constant is $\gamma X_d^2$.

$\square$

### A.2  PROOF OF LEMMA 4.1

*Proof.* Based on the definition of Lipschitz continunity, we have:

$$\begin{aligned}
\|F(\theta) - F(\theta')\|_2 &= \|(\theta - \eta\nabla\mathcal{F}(\theta)) - (\theta' - \eta\nabla\mathcal{F}(\theta'))\|_2 \\
&\leq \left\|\left(\mathbf{I} - \eta\cdot\frac{\nabla\mathcal{F}(\theta) - \nabla\mathcal{F}(\theta')}{\theta - \theta'}\right)(\theta - \theta')\right\|_2 \\
&= \|\mathbf{I} - \eta\nabla^2\mathcal{F}(\widetilde{\theta})\|_2\|\theta - \theta'\|_2
\end{aligned}$$

where $\widetilde{\theta}$ is between $\theta$ and $\theta'$. Based on the smoothness of $\mathcal{F}$ and the assumption, we have $\omega\mathbf{I} \leq \nabla^2\mathcal{F}(\widetilde{\theta}) \leq \beta\mathbf{I}$. Therefore, $\|\mathbf{I} - \eta\nabla^2\mathcal{F}(\widetilde{\theta})\|_2 \leq \max\{|1 - \eta\omega|, |1 - \eta\beta|\}$, which is the bound of the Lipschitz constant.

$\square$

### A.3  PROOF OF LEMMA 4.2

*Proof.* Let $\widetilde{\theta}_1 = Prox_{\eta,r}(\theta_1)$ and $\widetilde{\theta}_2 = Prox_{\eta,r}(\theta_2)$, we then have the following inequalities based on the optimality:

$$\begin{aligned}
r(\widetilde{\theta}_1) + \frac{1}{2\eta}\left\|\widetilde{\theta}_1 - \theta_1\right\|_2^2 &\leq r\left(\frac{\widetilde{\theta}_1 + \widetilde{\theta}_2}{2}\right) + \frac{1}{2\eta}\left\|\frac{\widetilde{\theta}_1 + \widetilde{\theta}_2}{2} - \theta_1\right\|_2^2 \\
r(\widetilde{\theta}_2) + \frac{1}{2\eta}\left\|\widetilde{\theta}_2 - \theta_2\right\|_2^2 &\leq r\left(\frac{\widetilde{\theta}_1 + \widetilde{\theta}_2}{2}\right) + \frac{1}{2\eta}\left\|\frac{\widetilde{\theta}_1 + \widetilde{\theta}_2}{2} - \theta_2\right\|_2^2
\end{aligned} \tag{12}$$

Based on the convexity of the function $r$, we have $2r\left(\frac{\widetilde{\theta}_1 + \widetilde{\theta}_2}{2}\right) \leq r(\widetilde{\theta}_1) + r(\widetilde{\theta}_2)$. Summing this inequality and the ones in (12), we obtain the following inequality.

$$\left\|\widetilde{\theta}_1 - \theta_1\right\|_2^2 + \left\|\widetilde{\theta}_2 - \theta_2\right\|_2^2 \leq \left\|\frac{\widetilde{\theta}_1 + \widetilde{\theta}_2}{2} - \theta_1\right\|_2^2 + \left\|\frac{\widetilde{\theta}_1 + \widetilde{\theta}_2}{2} - \theta_2\right\|_2^2$$

We simplify the inequality above and obtain $\|\widetilde{\theta}_2 - \widetilde{\theta}_1\|_2^2 \leq 2\langle\widetilde{\theta}_2 - \widetilde{\theta}_1, \theta_2 - \theta_1\rangle$. Since $\langle\widetilde{\theta}_2 - \widetilde{\theta}_1, \theta_2 - \theta_1\rangle \leq \|\widetilde{\theta}_2 - \widetilde{\theta}_1\|_2\|\theta_2 - \theta_1\|_2$, so we have $\|\widetilde{\theta}_2 - \widetilde{\theta}_1\|_2 \leq 2\|\theta_2 - \theta_1\|_2$. $\square$

### A.4  PROOF OF THEOREM A.2

For proof completeness, we first provide the formal definition of Log-Sobolev Inequality (LSI).

**Definition A.1.** *(Log-Sobolev Inequality Vempala and Wibisono (2019)) A distribution $\nu$ over $\mathbb{R}^d$ satisfies log-Sobolev inequality (LSI) with a constant $c$ if $\forall$ smooth function $g : \mathbb{R}^d \to \mathbb{R}$ with $\mathbb{E}_{\theta\sim\nu}[g^2(\theta)] < +\infty$,*

$$\mathbb{E}_{\theta\sim\nu}[g^2(\theta)\log(g^2(\theta))] - \mathbb{E}_{\theta\sim\nu}[g^2(\theta)]\log\mathbb{E}_{\theta\sim\nu}[g^2(\theta)] \leq \frac{2}{c}\mathbb{E}_{\theta\sim\nu}[\|\nabla g(\theta)\|_2^2] \tag{13}$$

Then, we go to the formal proof of Theorem A.2.

**Theorem A.2.** *(Rate of RDP) Let $\mu$, $\nu$ be two distributions on $\mathbb{R}^d$. $F, T : \mathbb{R}^d \to \mathbb{R}^d$ are measurable mappings. We use $p_t(\theta)$ and $p'_t(\theta)$ to represent the probability density functions of $F_{\#}(\mu) * \mathcal{N}(0, 2t \cdot o(\eta, k, j) \cdot \mathbb{I}_d)$ and $F_{\#}(\nu) * \mathcal{N}(0, 2t \cdot o(\eta, k, j) \cdot \mathbb{I}_d)$. In addition, we use $h_t(\theta)$ and $h'_t(\theta)$ to represent the probability desity function of $T_{\#}(F_{\#}(\mu) * \mathcal{N}(0, 2t \cdot o(\eta, k, j) \cdot \mathbb{I}_d))$ and $T_{\#}(F_{\#}(\nu) * \mathcal{N}(0, 2t \cdot o(\eta, k, j) \cdot \mathbb{I}_d))$. Furthermore, we use $P$ to represent the probability transition function by the mapping $T_{\#}$, i.e., $h_t(\theta) = P(p_t(\theta))$ and $h'_t(\theta) = P(p'_t(\theta))$ are composition functions. If $\mu$, $\nu$ satisfy log-sobolev inequality (LSI) with constant $c$, and if the mappings $F, T$ are $L_F, L_T$-Lipschitz continuous, $P$ is a linear function, then for any $\alpha > 1$, we have the following bound for the Rényi divergence of order $\alpha$ between $h_t(\theta)$ and $h'_t(\theta)$:*

$$\frac{\partial}{\partial t} R_\alpha(h_t(\theta) \| h'_t(\theta)) \le -2c_t \cdot o(\eta, k, j) \cdot \left( \frac{R_\alpha(h_t(\theta) \| h'_t(\theta))}{\alpha} + (\alpha - 1) \frac{\partial}{\partial \alpha} R_\alpha(h_t(\theta) \| h'_t(\theta)) \right) \tag{14}$$

*where $c_t = \left( \frac{L_F^2}{c} + 2t \cdot o(\eta, k, j) \right)^{-1} / L_T^2$.*

*Proof.* Denote $E_\alpha(h_t(\theta) \| h'_t(\theta)) = \int h'_t(\theta) \cdot \frac{h_t(\theta)^\alpha}{h'_t(\theta)^\alpha} d\theta$ to be the moment of the likehood ratio function, then

$$R_\alpha(h_t(\theta) \| h'_t(\theta)) = \frac{1}{\alpha - 1} \log E_\alpha(h_t(\theta) \| h'_t(\theta)) \tag{15}$$

We compute the rate of Rényi divergence with regard to $t$ as follows:

$$\begin{aligned} \frac{\partial R_\alpha h_t(\theta) \| h'_t(\theta)}{\partial t} &= \frac{1}{\alpha - 1} \log E_\alpha(h_t(\theta) \| h'_t(\theta)) \\ &= \frac{1}{(\alpha - 1) E_\alpha(h_t(\theta) \| h'_t(\theta))} \cdot \frac{\partial}{\partial t} \left( \int \frac{h_t(\theta)^\alpha}{h'_t(\theta)^{\alpha-1}} d\theta \right) \end{aligned} \tag{16}$$

By exchanging the order of derivative and integration since they are with respect to different variables, we have:

$$\begin{aligned} \frac{\partial R_\alpha(h_t(\theta) \| h'_t(\theta))}{\partial t} = &\frac{1}{(\alpha - 1) E_\alpha(h_t(\theta) \| h'_t(\theta))} \cdot \\ &\int \left( \alpha \cdot \frac{h_t(\theta)^{\alpha-1}}{h'_t(\theta)^{\alpha-1}} \cdot \frac{\partial h_t(\theta)}{\partial p_t(\theta)} \cdot \frac{\partial p_t(\theta)}{\partial t} \right. \\ &\left. - (\alpha - 1) \cdot \frac{h_t(\theta)^\alpha}{h'_t(\theta)^\alpha} \cdot \frac{\partial h'_t(\theta)}{\partial p'_t(\theta)} \cdot \frac{\partial p'_t(\theta)}{\partial t} \right) d\theta \end{aligned} \tag{17}$$

Since the $\mu * \mathcal{N}(0, 2t \cdot o(\eta, k, j) \cdot \mathbb{I}_d)$ and $\nu * \mathcal{N}(0, 2t \cdot o(\eta, k, j) \cdot \mathbb{I}_d)$ are heat flow at time $t \in [0, o(\eta, k, j)]$. Therefore $p_t(\theta)$ and $p'_t(\theta)$ satisfy the following Fokker-Planck equations (Kadanoff, 2000).

$$\frac{\partial p_t(\theta)}{\partial t} = o(\eta, k, j) \Delta p_t(\theta), \quad \frac{\partial p'_t(\theta)}{\partial t} = o(\eta, k, j) \Delta p'_t(\theta) \tag{18}$$

Then Equation (17) can be written as

$$\begin{aligned} \frac{\partial R_\alpha(h_t(\theta) \| h'_t(\theta))}{\partial t} = &\frac{o(\eta, k, j)}{(\alpha - 1) E_\alpha(h_t(\theta) \| h'_t(\theta))} \cdot \\ &\int \left( \alpha \cdot \frac{h_t(\theta)^{\alpha-1}}{h'_t(\theta)^{\alpha-1}} \cdot \frac{\partial h_t(\theta)}{\partial p_t(\theta)} \cdot \Delta p_t(\theta) \right. \\ &\left. - (\alpha - 1) \cdot \frac{h_t(\theta)^\alpha}{h'_t(\theta)^\alpha} \cdot \frac{\partial h'_t(\theta)}{\partial p'_t(\theta)} \cdot \Delta p'_t(\theta) \right) d\theta \end{aligned} \tag{19}$$

We apply Green's first identity to further simplify the equation above:

$$\int \alpha \cdot \frac{h_t(\theta)^{\alpha-1}}{h_t'(\theta)^{\alpha-1}} \cdot \frac{\partial h_t(\theta)}{\partial p_t(\theta)} \cdot \Delta p_t(\theta)d\theta = -\int \nabla \left( \alpha \cdot \frac{h_t(\theta)^{\alpha-1}}{h_t'(\theta)^{\alpha-1}} \cdot \frac{\partial h_t(\theta)}{\partial p_t(\theta)} \right) \cdot \nabla p_t(\theta)d\theta$$

$$= -\alpha \int \nabla \left( \frac{h_t(\theta)^{\alpha-1}}{h_t'(\theta)^{\alpha-1}} \right) \cdot \frac{\partial h_t(\theta)}{\partial p_t(\theta)} \cdot \nabla p_t(\theta)d\theta \qquad (20)$$

$$= -\alpha \int \nabla \left( \frac{h_t(\theta)^{\alpha-1}}{h_t'(\theta)^{\alpha-1}} \right) \cdot \nabla h_t(\theta)d\theta$$

To avoid confusion, all $\nabla$ operators represent the derivative with respect to $\theta$. Note that $h_t(\theta) = P(p_t(\theta))$ and $P$ is a linear function, so $\frac{\partial h_t(\theta)}{\partial p_t(\theta)}$ is a constant. This is why we can move the term $\frac{\partial h_t(\theta)}{\partial p_t(\theta)}$ outside the $\nabla$ operator.

Using the same technique, we can bound the second term of Equation (19):

$$\int -(\alpha-1) \cdot \frac{h_t(\theta)^{\alpha}}{h_t'(\theta)^{\alpha}} \cdot \frac{\partial h_t(\theta)}{\partial p_t(\theta)} \cdot \Delta p'(\theta)d\theta = (\alpha-1) \int \nabla \left( \frac{h_t(\theta)^{\alpha}}{h_t'(\theta)^{\alpha}} \right) \cdot \nabla h_t'(\theta)d\theta \qquad (21)$$

Plug Equation (20) and (21) into Equation (19), we have the following equation:

$$\frac{\partial R_\alpha(h_t(\theta)\|h_t'(\theta))}{\partial t} = \frac{\alpha \cdot o(\eta,k,j)}{E_\alpha(h_t(\theta)\|h_t'(\theta))} \cdot \left( \frac{1}{\alpha} \int \nabla \left( \frac{h_t(\theta)^{\alpha}}{h_t'(\theta)^{\alpha}} \right) \cdot \nabla h_t'(\theta)d\theta \right.$$

$$\left. - \frac{1}{\alpha-1} \int \nabla \left( \frac{h_t(\theta)^{\alpha-1}}{h_t'(\theta)^{\alpha-1}} \right) \cdot \nabla h_t(\theta)d\theta \right)$$

$$= \frac{\alpha \cdot o(\eta,k,j)}{E_\alpha(h_t(\theta)\|h_t'(\theta))} \left( \int \frac{h_t(\theta)^{\alpha-1}}{h_t'(\theta)^{\alpha-1}} \cdot \left\langle \nabla \left( \frac{h_t(\theta)}{h_t'(\theta)} \right), \nabla h_t'(\theta) \right\rangle d\theta \right.$$

$$\left. - \int \frac{h_t(\theta)^{\alpha-2}}{h_t'(\theta)^{\alpha-2}} \cdot \left\langle \nabla \left( \frac{h_t(\theta)}{h_t'(\theta)} \right), \nabla h_t(\theta) \right\rangle d\theta \right)$$

$$= -\frac{\alpha \cdot o(\eta,k,j)}{E_\alpha(h_t(\theta)\|h_t'(\theta))} \int \frac{h_t(\theta)^{\alpha-2}}{h_t'(\theta)^{\alpha-2}} \cdot \left\langle \nabla \left( \frac{h_t(\theta)}{h_t'(\theta)} \right), \nabla \left( \frac{h_t(\theta)}{h_t'(\theta)} \right) \right\rangle h_t'(\theta)d\theta$$

$$\overset{\text{def}}{=} -\alpha \cdot o(\eta,k,j) \cdot \frac{I_\alpha(h_t(\theta)\|h_t'(\theta))}{E_\alpha(h_t(\theta)\|h_t'(\theta))} \qquad (22)$$

where we define $I_\alpha(h_t(\theta)\|h_t'(\theta)) = \int \frac{h_t(\theta)^{\alpha-2}}{h_t'(\theta)^{\alpha-2}} \cdot \left\langle \nabla \left( \frac{h_t(\theta)}{h_t'(\theta)} \right), \nabla \left( \frac{h_t(\theta)}{h_t'(\theta)} \right) \right\rangle \cdot h_t'(\theta)d\theta = \mathbb{E}_{\theta \sim h_t'} \left( \frac{h_t(\theta)^{\alpha}}{h_t'(\theta)^{\alpha}} \left\| \nabla \log \frac{h_t(\theta)}{h_t'(\theta)} \right\|_2^2 \right)$.

Based on Vempala and Wibisono (2019, Lemma 16, Lemma 17), we can conclude $h_t(\theta)$ and $h_t'(\theta)$ satisfy log-Sobolev inequality with a constant $c_t = \left( \frac{L_F^2}{c} + 2t \cdot o(\eta,k,j) \right)^{-1} / L_T^2$. In this regard, we can utilize Ye and Shokri (2022, Lemma D.1) and Vempala and Wibisono (2019, Lemma 5) to bound $\frac{I_\alpha(h_t(\theta)\|h_t'(\theta))}{E_\alpha(h_t(\theta)\|h_t'(\theta))}$ as follows:

$$\frac{I_\alpha(h_t(\theta)\|h_t'(\theta))}{E_\alpha(h_t(\theta)\|h_t'(\theta))} \geq \frac{2c_t}{\alpha^2} \cdot R_\alpha(h_t(\theta)\|h_t'(\theta)) + \frac{2c_t}{\alpha^2} \cdot \alpha(\alpha-1) \frac{\partial}{\partial \alpha} R_\alpha(h_t(\theta)\|h_t'(\theta)) \qquad (23)$$

Combine Inequality (23) with Equation 22, we conclude the proof.

$\square$

The theorem A.2 bound the rate of Rényi divergence follows Ye and Shokri (2022); Vempala and Wibisono (2019). One assumption in Theorem A.2 is $P$ being a linear function. The function $P$ depicts the change of the probability density functions before and after the proximal operator.

For the three typical regularization function $r$ discussed previously, including no regularization, $l_2$ regularization and $l_1$ regularization, it is clear that all their corresponding $P$ functions are linear. In addition, the Lipschitz continuity of function $F$ and $T$ is guaranteed in Lemma 4.1 and Lemma 4.2, respectively. Therefore, we can conclude that the assumptions in Theorem A.2 are not restrictive.

Theorem A.2 considers the recursive privacy dynamics during one step of noisy mini-batch proximal gradient descent in (8). In the corollary 4.4, we apply the Theorem A.2 in the context of differential privacy. Under Assumption 4.3, we can then apply Theorem A.2 to the case of two neighboring datasets and obtain the following corollary.

## A.5   Proof of Theorem 4.5

Before proving the main theorem, we first calculate the LSI constant sequence in Algorithm 2.

**Lemma A.3.**  *(LSI constant sequence in Algorithm 2) For each layer's update scheme in Algorithm 2 with a batch size of $b$, if the update function $F(\theta) = \theta - \eta \nabla f(\theta)$ and the proximal operator $T(\theta) = Prox_{\eta,r}$ are Lipschitz continuous with Lipschitz constant $L_F$ and $L_T$, respectively, then the distribution of parameter $\theta_k^j$ in the $j$-th iteration of the $k$-th epoch satisfies $c_k^j$ log-Sobolev inequality (LSI) and the constant $c_k^j$ is calculated by:*

$$
c_k^j = \frac{1}{2\eta L_T^2} \left( \sum_{k'=0}^{k-1} \sum_{j'=0}^{n/b-1} o(\eta, k', j')(L_F L_T)^{2((k-k')(n/b)-j'+j-1)} \right.
$$
$$
\left. + \sum_{j'=0}^{j-1} o(\eta, k, j')(L_F L_T)^{2(j-j'-1)} \right)^{-1}
$$

*Proof.* Based on Definition A.1, the LSI constant of the Gaussian distribution $\mathcal{N}(0, 2t \cdot o(\eta, k, j))$ is $\frac{1}{2t \cdot o(\eta, k, j)}$.

Then, Using the LSI under Lipschitz mapping (Vempala and Wibisono, 2019, lemma 16) and Gaussian convolution (Vempala and Wibisono, 2019, lemma17), we have $\frac{1}{c_k^j} = \frac{L^2}{c_k^{j-1}} + 2\eta L_T^2 \cdot o(\eta, k, j-1)$ where $L = L_F L_T$ for notation simplicity. By applying this equation iteratively via $j$ and $k$, we obtain:

$$
\frac{1}{c_k^j} = \frac{L^{2j}}{c_k^0} + 2\eta L_T^2 \cdot \sum_{j'=0}^{j-1} o(\eta, k, j') L^{2(j-j'-1)}
$$
$$
= \frac{L^{2(k \cdot n/b+j)}}{c_0^0} + 2\eta L_T^2 \cdot \left( \sum_{k'=0}^{k-1} \sum_{j'=0}^{n/b-1} o(\eta, k', j') L^{2((k-k')(n/b)-j'+j-1)} \right.
$$
$$
\left. + \sum_{j'=0}^{j-1} o(\eta, k, j') L^{2(j-j'-1)} \right)
$$

Because the initialization is point distribution around $\theta_0$, $\theta_0^0$ satisfies the log-Sobolev inequality with constant $c_0^0 = \infty$, and the lemma is then proved.  $\square$

We now return to Theorem 4.5 and provide the complete proof below.

*Proof.* We denote $\theta_k^j$ and $\theta_k'^j$ as the intermediate parameters in the $j$-th iteration of the $k$-th epoch in Algorithm 1 when using two neighboring datasets $D$ and $D'$. In addition, we use $\varepsilon_k^j(\alpha) = R_\alpha(\theta_k^j || \theta_k'^j)$ to represent the Rényi divergence of order $\alpha$ between them. In this regard, it is clear that $\varepsilon_0^0(\alpha) = 0$.

Without the loss of generality, we assume that the only different data point is in the $j_0$-th batch. Therefore, the privacy bound after $K$ epochs can be decomposed into three parts by using 4.4: 1)

the first $j_0 - 1$ mini-batch updates in the first epoch; 2) the remaining mini-batch updates in the first epoch; 3) the rest epochs.

In the first stage, we have $\forall j \in \{0, 1, \ldots, j_0 - 1\}$ and $\forall \alpha > 1, \varepsilon_0^j(\alpha) = 0$ based on Lemma 4.4.

In the second stage, we have $\forall \alpha, \frac{\varepsilon_0^{j_0}(\alpha)}{\alpha} \leq \frac{\eta S_g^2}{4b^2 \cdot o(\eta, 0, j_0)}$. Then we bound the privacy loss in the end of the first epoch by the following inequality. Note that, this inequality is applicable for any $\alpha > 1$.

$$\forall \alpha > 1, \quad \frac{\varepsilon_0^{n/b-1}(\alpha)}{\alpha} \leq \frac{\eta S_g^2}{4b^2 \cdot o(\eta, 0, j_0)} \prod_{j=j_0+1}^{n/b-1} \left(1 + \frac{c_0^j \cdot 2\eta \cdot o(\eta, 0, j)}{L_F^2}\right)^{-1/L_T^2} \tag{24}$$

In the third stage, we first consider the second epoch, especially the $j_0$-th iteration and the last iteration. Based on Lemma 4.4, we have the following inequalities:

$$\frac{\varepsilon_1^{j_0}(\alpha)}{\alpha} \leq \frac{\varepsilon_1^{-1}(\alpha')}{\alpha'} \prod_{j=0}^{j_0-1} \left(1 + \frac{c_1^j \cdot 2\eta \cdot o(\eta, 1, j)}{L_F^2}\right)^{-1/L_T^2} + \frac{\eta S_g^2}{4b^2 \cdot o(\eta, 1, j_0)}$$

$$\frac{\varepsilon_1^{n/b-1}(\alpha)}{\alpha} \leq \frac{\varepsilon_1^{j_0}(\alpha'')}{\alpha''} \prod_{j=j_0+1}^{n/b-1} \left(1 + \frac{c_1^j \cdot 2\eta \cdot o(\eta, 1, j)}{L_F^2}\right)^{-1/L_T^2}$$

$$\leq \left(\frac{\varepsilon_1^{-1}(\alpha''')}{\alpha'''} \prod_{j=0}^{j_0-1} \left(1 + \frac{c_1^j \cdot 2\eta \cdot o(\eta, 1, j)}{L_F^2}\right)^{-1/L_T^2} + \frac{\eta S_g^2}{4b^2 \cdot o(\eta, 1, j_0)}\right) \tag{25}$$

$$\cdot \prod_{j=j_0+1}^{n/b-1} \left(1 + \frac{c_1^j \cdot 2\eta \cdot o(\eta, 1, j)}{L_F^2}\right)^{-1/L_T^2}$$

Here, we use $\varepsilon_1^{-1}$ to represent the privacy bound before the first iteration of the second epoch. It is clear that $\varepsilon_1^{-1} = \varepsilon_0^{n/b-1}$. In addition, $\alpha'$, $\alpha''$ and $\alpha'''$ are the $j_0$, $(n/b - 1 - j_0)$ and $(n/b - 1)$ fold mapping value of $\alpha$ under the repeated mapping $\alpha \leftarrow (\alpha - 1) \cdot \left(1 + \frac{c_1^j \cdot 2\eta \cdot o(\eta, 1, j)}{L_F^2}\right)^{-1/L_T^2}$, respectively.

Considering inequality (24) holds for any $\alpha > 1$ and the fact $\alpha''' > 1$, we can then plug inequality (24) into inequality (25) and obtain the following inequality:

$$\frac{\varepsilon_1^{n/b-1}(\alpha)}{\alpha} \leq \frac{\eta S_g^2}{4b^2 \cdot o(\eta, 0, j_0)} \prod_{j=j_0+1}^{n/b-1} \left(1 + \frac{c_0^j \cdot 2\eta \cdot o(\eta, 0, j)}{L_F^2}\right)^{\frac{-1}{L_T^2}}$$

$$\prod_{j=0, j\neq j_0}^{n/b-1} \left(1 + \frac{c_1^j \cdot 2\eta \cdot o(\eta, 1, j)}{L_F^2}\right)^{\frac{-1}{L_T^2}} + \frac{\eta S_g^2}{4b^2 \cdot o(\eta, 1, j_0)} \prod_{j=j_0+1}^{n/b-1} \left(1 + \frac{c_1^j \cdot 2\eta \cdot o(\eta, 1, j)}{L_F^2}\right)^{\frac{-1}{L_T^2}} \tag{26}$$

By recursively applying Equation (26), we can obtain the privacy bound of the whole training phase. For notation simplicity, we use $\Phi(k_1, k_2)$ to denote the RDP's decay rate between the $k_1$-th epoch and the $k_2$-th epoch.

$$\Phi(k_1, k_2) = \prod_{k=k_1}^{k_2-1} \prod_{j=0, j\neq j_0}^{n/b-1} \left(1 + \frac{c_k^j \cdot 2\eta \cdot o(\eta, k, j)}{L_F^2}\right)^{-1/L_T^2} \tag{27}$$

In Lemma A.3 we have $\frac{1}{c_k^{j+1}} = \frac{L_F^2 L_T^2}{c_k^j} + 2\eta L_T^2 \cdot o(\eta, k, j)$ based on the LSI under Lipschitz mapping (Vempala and Wibisono, 2019, Lemma 16) and Gaussian convolution (Vempala and

Wibisono, 2019, Lemma 17), so $1 + \frac{c_k^j \cdot 2\eta \cdot o(\eta,k,j)}{L_F^2} = \frac{c_k^j}{c_k^{j+1}} \cdot \frac{1}{L_F^2 L_T^2}$. Therefore, Equation (27) can be further simplified as follows:

$$\Phi(k_1, k_2) = \left( \frac{c_{k_1}^0}{c_{k_2-1}^{n/b-1}} \left( \frac{1}{L_F^2 L_T^2} \right)^{(n/b-1)(k_2-k_1)} \prod_{k=k_1}^{k_2-1} \frac{c_k^{j_0+1}}{c_k^{j_0}} \right)^{-1/L_T^2} \tag{28}$$

Therefore, we can obtain the privacy bound after $K$ epochs as follows:

$$\frac{\varepsilon_K^{n/b-1}(\alpha)}{\alpha} \leq \sum_{k=0}^{K-1} \frac{\eta S_g^2}{4b^2 \cdot o(\eta,k,j_0)} \cdot \Phi(k+1,K) \cdot \prod_{j=j_0+1}^{n/b-1} \left( 1 + \frac{c_k^j \cdot 2\eta \cdot o(\eta,k,j)}{L_F^2} \right)^{-1/L_T^2}$$

$$= \sum_{k=0}^{K-1} \frac{\eta S_g^2}{4b^2 \cdot o(\eta,k,j_0)} \left( \frac{c_k^{j_0+1}}{c_K^{n/b-1}} \left( \frac{1}{L_F^2 L_T^2} \right)^{(n/b-1)(K-k)-j_0} \prod_{l=k}^{K} \frac{c_l^{j_0+1}}{c_l^{j_0}} \right)^{-1/L_T^2} \tag{29}$$

Note that the right-hand side of inequality (29) depends on $j_0$ as well. We now explicitly rewrite $\varepsilon_K^{n/b-1}(\alpha)$ as $\varepsilon_K^{n/b-1}(\alpha, j_0)$ and bound the value of $\varepsilon_K(\alpha)$ in a similar way to Ye and Shokri (2022, Theorem 4.2):

$$\varepsilon_K(\alpha) = \mathbb{E}_{j_0} \varepsilon_K^{n/b-1}(\alpha, j_0) = \frac{1}{\alpha-1} \log e^{(\alpha-1)\mathbb{E}_{j_0} \varepsilon_K^{n/b-1}}$$

$$\leq \frac{1}{\alpha-1} \log \left( \mathbb{E}_{j_0} e^{(\alpha-1)\varepsilon_K^{n/b-1}} \right) = \frac{1}{\alpha-1} \log \left( \sum_{j_0=0}^{n/b-1} \frac{b}{n} \cdot e^{(\alpha-1)\varepsilon_K^{n/b-1}(\alpha,j_0)} \right) \tag{30}$$

$\square$

# B NUMERICAL EXPERIMENTS

In this section, we first use several datasets to verify the utility of our algorithm. Then, we further shows that the adaptive noise could provide better privacy-utility trade-off. To accelerate the algorithm, we use a variant of the Algorithm 2 demonstrated in the Appendix C.

## B.1 UTILITY VERIFICATION

In this section, we apply our algorithms on MNIST and Adults, which are widely used in membership inference attacks, to show the validation of our algorithm.

We employ a four-layer multilayer perceptron (MLP) model, where each layer consists of 1200 neurons. The activation function is ReLU, and we use the squared loss as the loss function. To reduce the variance, we use data augmentation to double the original dataset and set the batch size to 60000. Furthermore, we set the layer-wise Lipschitz constant $\rho = 1$ and the step size $\eta = 0.99$. Since this experiment primarily demonstrates the effectiveness of our algorithm, we provide a **fixed differential privacy setting** rather than different initial noise, which will be discussed in the next subsection. The experiment chooses $\alpha = 100$ and adds calibrated noise of per epoch, resulting in a final noise variance of $o(\eta, K, j) = 0.001$. We implement our algorithm with above setting on both MNIST dataset and Adult dataset. Based on Theorem 4.5. Note that the two dataset shares nearly the same RDP loss is because they shares the same setting except $b$. The experiment demonstrates that our algorithm can converge with a small privacy loss. We also implement our algorithm on the (convolution neural network) CNN. The implementation trick is demonstrated in Appendix C.1.

We have to emphasize that we cannot directly compare the privacy loss between DP-SBCD under the hidden state assumption and the DP-SGD with the DP-SGD under the composition theorem because that the two algorithms the assumption are totally different, and they also holds totally different settings. However, in order to provide a intuitively comparison between two algorithm, we also compare the utility between the two algorithm. We use the FastDP library Bu et al. (2023) with the auto-clipping mode, which normalize the gradient and add calibrate noise. Bu et al. (2024) shows

| Dataset | Avg Acc (DP-SBCD) | Architecture | $(\alpha, \epsilon)$ RDP Loss |
|---------|-------------------|--------------|-------------------------------|
| MNIST | 94.63($\pm$0.68) | MLP | $(100, 0.0404)$ |
| MNIST | 96.36($\pm$0.42) | LeNet | $(100, 0.0506)$ |
| Fashion-MNIST | 82.82($\pm$0.80) | MLP | $(100, 0.0404)$ |
| Fashion-MNIST | 79.21($\pm$0.97) | LeNet | $(100, 0.0506)$ |
| Adults | 83.57($\pm$0.82) | MLP | $(100, 0.0404)$ |

Table 1: The accuracy(in % under 95% confidential interval) for DP-SBCD with hidden state assumption

that this mode could outperforms or matches the state-of-the-art. The auto clipping mechanism also simplify the super parameter choose. For vanilla DP-SGD, we need to carefully choose clipping threshold as well as learning rate, which make it more difficult to balance the privacy loss and utility The epoch is 50 because the number of epoch is necessary for the privacy accountant of the DP-SGD algorithm. Moreover, the utility may be even worse when epoch is larger because it will add larger calibrated noise to the training gradient. 2. Moreover, the learning rate is crucial for DP-SGD's utility and privacy loss accountant. In our experiment, we choose different learning rate for different architecture and dataset.

| Dataset | Avg Acc (DP-SBCD) | Avg Acc (DP-SGD) | Architecture | $(\alpha, \epsilon)$ RDP Loss |
|---------|-------------------|------------------|--------------|-------------------------------|
| MNIST | 94.63($\pm$0.68) | 80.09($\pm$0.55),lr=0.1 | MLP | $(100, 0.0404)$ |
| MNIST | 96.36($\pm$0.42) | 81.80($\pm$1.43),lr=0.5 | LeNet | $(100, 0.0506)$ |
| Fashion-MNIST | 82.82($\pm$0.80) | 74.74($\pm$0.43),lr=0.05 | MLP | $(100, 0.0404)$ |
| Fashion-MNIST | 79.21($\pm$0.97) | 70.84($\pm$1.28),lr=0.5 | LeNet | $(100, 0.0506)$ |
| Adults | 83.57($\pm$0.82) | 75.86($\pm$0.63),lr=0.01 | MLP | $(100, 0.0404)$ |

Table 2: The accuracy(in % under 95% confidential interval) for DP-SBCD and DP-SGD where epoch is 50

## B.2 UTILITY-PRIVACY TRADE-OFF

In this section, we run numerical simulations in this section to investigate the model's utility and privacy loss in different training phases when we use different distributions to sample calibrated noise.

The Madelon dataset was originally introduced as a challenging classification problem in the NIPS 2003 feature selection challenge (Guyon et al., 2004). This synthetic dataset comprises 6000 instances, each with 20 features and belonging to one of the five classes. To ensure an unbiased evaluation, we divided the dataset into a training set, accounting for 80% of the data, and a testing set, containing the remaining 20%. The batch size is 960.

We employ a smaller with four layers, each of which has 200 neurons. The activation function is the commonly used ReLU function, and we use squared loss as the loss function. In addition, we set the layerwise Lipschitz constant $\rho = 0.1$ and the step size $\eta = 0.99$. To guarantee privacy, we applied Theorem 4.5 and employed the privacy loss calculation with $\alpha = 100$. All the experiments can be efficiently executed on a single NVIDIA RTX 5000 Ada GPU.

The experiment compares the algorithm's utility under different noise strategies: the constant strategy with $o(\eta, k, j) = 0.014$ and the decrease strategy with a linear decay rate of $0.00075$ per epoch and final noise variance $o(\eta, K, j) = 0.01$. The decrease strategy is designed in a manner so that the privacy loss of both strategies will be approximately the same for the same number of epochs $K$. Based on the aforementioned setup, we train the model 40 times where the total number of epochs $K$ varies from 10 to 50. We run the whole experiment 3 times and report both the average performance and the standard deviation. The results are demonstrated in Figure 3 and Table 3.

The experiment results validate the effectiveness of Algorithm 2. What's more, with proper settings of adaptive calibrated noise, the algorithm can demonstrate a better trade-off between the model's utility and privacy. For the examples in Table 3, we see higher utilities and lower privacy loss when we use adaptive calibrated noise. In Figure 3, we see the curve of adaptive calibrated noise above one of its counterparts in most cases.

Table 3: The accuracy(in % under 95% confidential interval) and the privacy loss of Algorithm 2 when we use different noise strategies and train the model for different numbers of epochs. D means noise decrease scenario and C means noise constant scenario

| EPOCH | NOISE | PRIVACY LOSS | AVG ACC. |
|---|---|---|---|
| 10 | D | 0.040466 | 35.08($\pm$5.99) |
| 20 | D | 0.040538 | 68.57($\pm$9.58) |
| 30 | D | 0.040652 | 89.78($\pm$1.92) |
| 40 | D | 0.040881 | 94.49($\pm$0.67) |
| 10 | C | 0.040466 | 23.95($\pm$1.94) |
| 20 | C | 0.040539 | 71.98($\pm$3.72) |
| 30 | C | 0.040655 | 85.03($\pm$3.21) |
| 40 | C | 0.040881 | 90.30($\pm$2.13) |

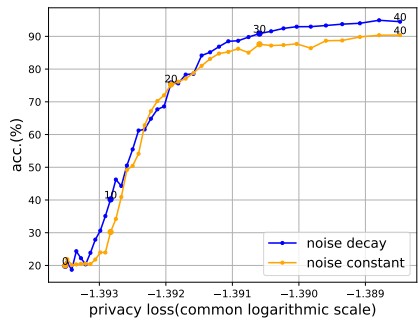

Figure 3: Relationship between model's utility and privacy loss under different noise strategies. For each noise strategy, we run Algorithm 2 for different numbers of epochs to plot the curve. The points are plotted in the order of epochs and we annotate some points aligned with the Table 3

## C  THE VARIANT OF ALGORITHM 2 USED IN THE EXPERIMENTS

In our experiment, we scale down the update scheme and employ weight decay to expedite the convergence rate. These techniques are aligned with our differential privacy analysis and do not increase privacy loss. We derive the algorithm in this section, omitting the subscript $d$ for simplicity.

For each iteration $i$, we have:

$$
\begin{aligned}
\theta^{(i+1)} &= \theta^{(i)} - \nabla_\theta \mathcal{F}(\theta) + \mathcal{N}(0, 2\eta \cdot o(\eta, k, j)\, \mathbf{I}) \\
&= \theta^{(i)} + \eta \left( U^T x - \theta^{(i)} x^T x \right) + \mathcal{N}(0, 2\eta \cdot o(\eta, k, j)\, \mathbf{I}) \\
&= \theta^{(i)}(\mathbf{I} - \eta x^T x) + \eta U^T x + \mathcal{N}(0, 2\eta \cdot o(\eta, k, j)\, \mathbf{I})
\end{aligned}
\tag{31}
$$

where $\nabla_w \mathcal{F}(\theta) = -(U^T x - \theta^{(i)} x^T x)$. We could regard the $\mathbf{I} - \eta x^T x)^{-1} \leq 1$ as a weigh decay term. In privacy loss analysis, we could ignore the weight decay term and regard the $L_T = 1$. Then, we have:

$$
\theta^{(i+1)} = \theta^{(i)} - \eta U^T x + \mathcal{N}(0, 2\eta \cdot o(\eta, k, j)\, \mathbf{I})
\tag{32}
$$

Since $(\mathbf{I} + \eta x^T x)^{-1} \leq 1$, it does not increase the privacy loss. Subsequently, we scale down the entire update scheme:

$$
\theta^{(i+1)} = (\theta^{(i)} - \eta U^T x + \mathcal{N}(0, 2\eta \cdot o(\eta, k, j)\, \mathbf{I})) \cdot (\mathbf{I} + \eta x^T x)^{-1}
\tag{33}
$$

Given that $(\mathbf{I} + \eta x^T x)^{-1} \leq 1$, the privacy loss remains unaffected. The rationale for transforming the original update scheme to (33) is that it is the close form of the original sub-problem:

$$
\theta_d \leftarrow \arg\min_{\theta_d'} \langle \mathcal{F}(\theta), \theta' - \theta \rangle + \frac{1}{2\eta} \|\theta' - \theta\|_F^2
\tag{34}
$$

In practice, employing the update schemes (33) is more convenient because the linear approximation form (31) typically requires some iterations to reach the optimal value of each sub-problem. However, the linear approximation format is more advantageous in theoretical analysis.

### C.1 Implementation on Convolutional Neural Network

The algorithm 2 can be also applied to CNN since the convolution operator can also be regard as a linear operations. More specifically, we could use the image to column algorithm to convert the $\theta * \boldsymbol{x}_d$ to $\boldsymbol{x}_d \theta$. Moreover, in solving the sub-problem $\mathcal{F}(\boldsymbol{x})$, we need to convert $\theta * \boldsymbol{x}_d$ to $\theta \boldsymbol{x}_d$ where $\boldsymbol{x}_d$ need to be flatten as a vector and generate the Topelitz matrix. We take a $3 \times 3$ input $x$ with a $2 \times 2$ kernel $w$, with stride equals 1 for example.

$$\begin{bmatrix} x_{11} & x_{12} & x_{21} & x_{22} \\ x_{12} & x_{13} & x_{22} & x_{23} \\ x_{21} & x_{22} & x_{31} & x_{32} \\ x_{22} & x_{23} & x_{32} & x_{33} \end{bmatrix} * \begin{bmatrix} w_{11} \\ w_{12} \\ w_{21} \\ w_{22} \end{bmatrix}$$

$$\begin{bmatrix} w_{11} & w_{12} & 0 & w_{21} & w_{22} & 0 & 0 & 0 & 0 \\ 0 & w_{11} & w_{12} & 0 & w_{21} & w_{22} & 0 & 0 & 0 \\ 0 & 0 & 0 & w_{11} & w_{12} & 0 & w_{21} & w_{22} & 0 \\ 0 & w_{11} & w_{12} & 0 & w_{21} & w_{22} & 0 & 0 & 0 \end{bmatrix} * \begin{bmatrix} x_{11} \\ x_{12} \\ x_{13} \\ x_{21} \\ x_{22} \\ x_{23} \\ x_{31} \\ x_{32} \\ x_{33} \end{bmatrix}$$

## D PRIVACY BOUND COMPARISON WITH YE AND SHOKRI (2022)

In this section, we intend to provide a intuitive comparison between our privacy loss accountant bound (Theorem 4.5) and (Ye and Shokri, 2022, Corollary 5.3).

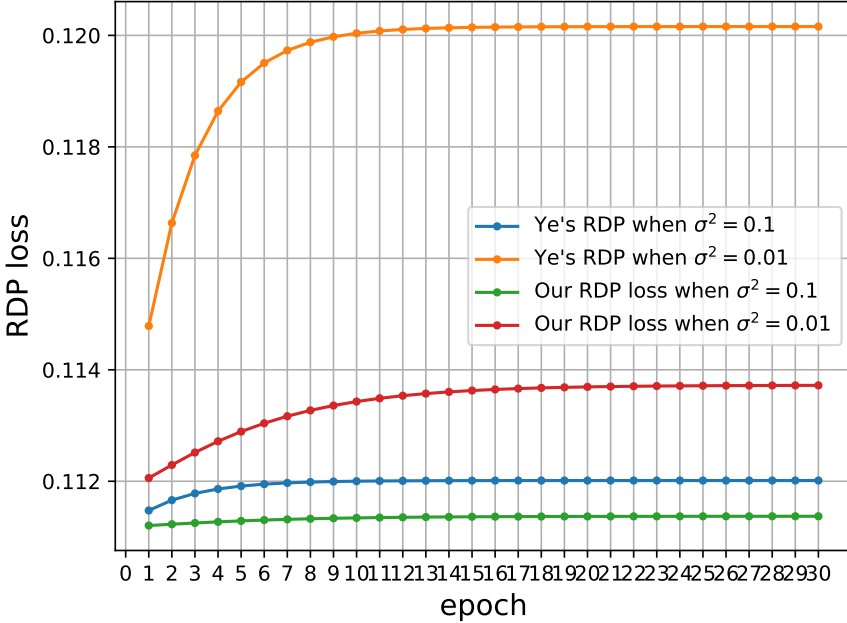

Figure 4: RDP loss comparison where common hyper-parameter settings are: learning rate $\eta = 0.1$, Lipschitz constant $L_F = 0.9$, batch size $b = 60$, data set $n = 240$, sensitivity $S_g = 1$, RDP order $\alpha = 10q$

The Figure 4 illustrates the privacy loss for a strongly convex problem with different calibrated noise for our privacy loss accountant method and Ye and Shokri (2022)'s method. It is evident that our algorithm provide a tighter privacy loss bound than Ye and Shokri (2022). The enhanced tightness of our bound is attributable to the comprehensive consideration of each epoch's privacy loss decreasing phase. Specifically, The Corollary 4.4 elucidates that when $i_0 \notin B_k^j$, the privacy loss will decrease exponentially. Ye and Shokri (2022)'s work approximate the overall privacy loss by overlook part of the decreasing phase in each epoch. However, when using adaptive noise, the calibrated noise can be modified in each iteration, so we carefully derive each iteration's privacy loss in proving Theorem 4.5 and yield a tighter bound.

