# OpenReview forum: "Differentially Private Network Training under Hidden State Assumption"
_ICLR.cc/2025/Conference — Submitted to ICLR 2025_

### Official Review · Reviewer_Hu7P · 2024-10-29

**Soundness:** 2
**Presentation:** 2
**Contribution:** 2
**Rating:** 5
**Confidence:** 3

**Summary:**

The paper proposes a method for training multi-layer neural networks with differential privacy (DP) guarantees in the hidden-state model of DP. This is obtained using a convex dual formulation of the network [(see Zeng et al., 2019, for non-DP literature)](http://proceedings.mlr.press/v97/zeng19a/zeng19a.pdf) and by using the privacy amplification by iteration analysis (Feldman et al., 2018). I see that the contribution is two-fold: a) introduce the convex reformulation for training multi-layer networks using the privacy amplification by iteraiton analysis and b) tailoring and improving the RDP analysis by Ye and Shokri (2022). Experimental results given in the appendix illustrate the effectiveness of the approach.

**Strengths:**

- I think the idea is nice. Although this convex reformulation (Eq. 3 in the paper) has not become very popular in the non-DP literature and it is unclear to me how well it compares to SGD training of non-convex formulations (and whether it can escape local minima etc.), I think it is interesting to study whether it could be beneficial in DP training. Essentially, all the DP updates are GD steps to convex subproblems so the privacy amplification by iteration analysis seems to fit well.

**Weaknesses:**

I am not sure about the soundness of the technical part of the paper, i.e., the RDP analysis for the privacy amplification by iteration. Let me explain in more detail:

- The analysis is essentially following the steps of [Ye and Shokri, 2022](https://proceedings.neurips.cc/paper_files/paper/2022/file/04b42392f9a3a16aea012395359b8148-Paper-Conference.pdf). First, the 2-page proof of Theorem A.2 given in the Appendix is a step-by-step copy of the proof of [Lemma 3.1 , Appendix of Ye and Shokri, 2022](https://proceedings.neurips.cc/paper_files/paper/2022/file/04b42392f9a3a16aea012395359b8148-Supplemental-Conference.pdf). While it is ok to repeat the steps, I think it should be mentioned that this is essentially the same proof.

- Second, the Corollary 4.4 also seems to be very similar to  [Lemma 3.2, Ye and Shokri, 2022](https://proceedings.neurips.cc/paper_files/paper/2022/file/04b42392f9a3a16aea012395359b8148-Paper-Conference.pdf). Here is the first point where the analysis deviates: instead of -1 there is $-1/L_T^2$ in that exponent. However, I cannot find the proof for Corollary 4.4 anywhere in the appendix.

- The rest of the analysis seems to also follow the steps of Ye and Shokri. I see that the only difference is that instead of having the strong convexity parameter $\lambda$ used in the analysis by Ye and Shokri (L2-regularization) you have the Lipschitz constant $L_T$ of the proximal operator. I think there is an inverse relation between $\lambda$ and $L_T$. It remains unclear to me, whether this analysis is really making improvement to the analysis by Ye and Shokri, since here $\lambda$ is hidden in $L_T$ and otherwise similar steps are carried out in the analysis. I think there should be some comparison (e.g., numerical) between these two analyses, it would be easy to carry out. There are some numerical results illustrating the RDP bounds in Figure 2, but the RDP order is not reported nor the Lipschitz constant $L_T$.  I think it should be indicated clearly where you make improvements compared to the analysis by Ye and Shokri.

- I think the experimental comparisons fall short. Those reported RDP parameters of Table 2 would correspond to approx. $(0.1,10^{-5})$-DP, and I find it hard to believe that DP-SGD for MNIST dataset would then give 10\% test accuracy. You say that DP-SGD and privacy amplification by iteration analyses are incomparable, though I think the comparison is ok since DP-SGD guarantees are valid also in the hidden state model. Also, using the analysis by Ye and Shokri for training a logistic regression, it is very difficult to beat the privacy-utility trade-off of DP-SGD, so looking at these positive results, it seems to me that the RDP bounds have to be much lower than those of DP-SGD. And this again raises the question where do the improvement in the analysis really come from (see above).

Small remark:

- Also the proof of Lemma 4.1 in the appendix can already be found in the appendix of [Feldman et al., 2018 (Proposition 18)](https://arxiv.org/pdf/1808.06651).


Due to the above reasons (e.g., a missing proof of a central lemma in the analysis), I think the paper is not yet mature for publication in its current form, though I think the idea is interesting and worth studying. I am ready to reconsider my score after the rebuttals.

**Questions:**

- Where does the exponent $-1/L_T^2$ in Corollary 4.4 come from ? How to prove Corollary 4.4 ?

- How would the RDP bounds compare to the RDP bounds by Ye and Shokri if you just consider a simple strongy convex model (e.g., logistic regression with L2-regularization) ? Where do the improvements then really come from? What is the relation of $\lambda$ and $L_T$ ?

---

> ### Author Response · Authors · 2024-11-20
> **Response for Weakness 1,2,3**
>
> Thanks for your detailed reading and problems, we are willing to answer it one by one:
>
> **1. The proof of Theorem A.2**
> The proof of Theorem A.2 is not a step-by-step copy, although we have mentioned that it is an **extension** of Ye and Shokri’s Lemma 3.1 in line 314.
> For Theorem 4.2, we consider the proximal gradient descent with adaptive calibrated noise. Therefore, we have **two pushforward functions** and **per iteration** adaptive Gaussian noise $o(\eta,k,j)$(line 705). More specifically, although we follow the basic framework as mentioned in line 314, the proof details are different from Equation (17). For the integrity of the work, we feel it necessary to put the full proof of Theorem A.2 in the appendix.
>
> **2. The proof of Corollary 4.4 **
>
> The proof of Corollary 4.4  basically follows the proof of Ye and Shokri (2022, Lemma 3.2), so we omit this part. We realize that in the original text, we inadvertently overlooked the explanation for the omission of corollary 4.4’s proof. We have rectified this problem in the **updated version line 325**.
>
> **3. The difference between Ye and Shokri in Corollary 5.3 and the order of RDP and the value of $L_T$ in Figure 2.**
>
> This answer elucidates the difference between the parameter $\lambda$ as delineated in Ye and Shokri, and our paper's $L_T$. The disparities in privacy loss analysis between our works and Ye and Shokri’s work that reviewer concerns will be answered in the response to the next question (4b), where corresponding numerical comparison is also presented.
>
> First, the effect of $\lambda$ in Ye & Shokri and $L_T$ in our work is different. $\lambda > 0$ is **necessary** in the framework of Ye & Shokri and they use $\lambda > 0$ as the coefficient of the $l_2$ regularization term to ensure that the overall loss objective function is strongly convex (See Equation (7) in Ye & Shokri). By contrast, the regularization term in our framework is **optional**. When there is no regularization, we can simply let $L_T = 1$ and our method still works. In addition, our framework supports any regularization term, as long as it is convex (in contrast to strongly-convex required in Ye & Shokri), which means our framework includes more regularization schemes such as lasso. Fundamentally, we can obtain feasible privacy loss under the hidden state assumption because the update rules mapping the old parameters to the new parameters (e.g. function $T$ and $F$ in Equation (9)) have bounded Lipschitz constants. **Such bounded Lipschitz constants are achieved in different ways in Ye & Shokri and our framework**. In Ye & Shokri, it is achieved by introducing an $l_2$ regularization scheme to make the loss function of regularized logistic regression strongly convex (See Equation (7) in Ye & Shokri). By contrast, we decompose deep neural network training into several sub-problems and each of these sub-problems have a convex loss function, this guarantees the bounded Lipschitz constant for the update rule (See Lemma 4.1 of our work which bounds $L_F$). The regularization scheme is optional and more generic in our framework. We discuss the regularization terms in line 304 to 308 to demonstrate our framework is generally applicable. In a nutshell, **both Ye & Shokri and our work manage to bound the Lipschitz constant of the update rule so that we can have a tight privacy loss estimation under the hidden state assumption, but the techniques used in our work to achieve Lipschitz bound are totally different from Ye & Shokri**
>
>
> In Figure 2, the RDP order is $\alpha = 100$ and  $L_T=1$. We clarify this in our revised manuscript. This again reflects that our framework is more generic regarding the regularization scheme. Our framework supports no regularization and any convex regularization (the value of $L_T$ for some examples are discussed in the paragraph from line 304 to 308), while Ye & Shokri requires an $l_2$ regularization term.

---

> > ### Author Response · Authors · 2024-11-20
> > **Response for Weakness 4,5 and Questions**
> >
> > **4. As for the experiment part:**
> >
> > **a. Why does DP-SGD have only 10% test accuracy?**
> >
> > In our experiment, in order to guarantee that DP-SGD for MNIST has the same privacy loss ($(100,0,04)$ RDP loss) and we can only use composition theorem to derive a loose privacy guarantee, we need to add extra calibrated noise to the DP-SGD algorithm. Moreover, we realize that DP-SGD also needs a smaller learning rate to prevent divergence while we fix the total number of epochs as $50$ in Table 2. All these conditions lead to a slower convergence rate, we observe that DP-SGD will eventually converge over a greater number of epochs. In addition, it is noteworthy that DP-SGD can achieve much better utility on MNIST when using LeNet compared with MLP.
> >
> > We believe it also shows the reason why we cannot directly compare the DP-SBCD and DP-SGD under different assumptions. For hidden state assumption, the convergence rate is also important, which is also indicated by Ye and Shorki, but for two different kinds of algorithms and assumptions, the empirical convergence rate can also be quite different under different settings such as the learning rate.
> >
> > **(b)Where do we make improvements compared to the analysis by Ye and Shokri?**
> >
> > **The key difference is the derivation of privacy loss in Theorem 4.5.** We provide a brief explanation in line 414 to 417 and we would love to provide further clarification.
> >
> > Ye & Shokri’s work  focuses on providing a theoretical guarantee that the privacy loss could converge under the hidden state assumption. Therefore, in order to derive a simplified result, Ye and Shorkri’s paper has some approximations in deriving their Theorem 4.2.  For example, they approximate the bound $\varepsilon_K(\alpha,j_0)$ by part of iterations in one epoch(Equation(101) in their work).
> >
> > However, when the calibrated noise is adaptive, **we need to carefully track the privacy loss among each iteration**. Using part of iterations privacy loss to approximate the whole epoch’s privacy loss, as indicated in Ye and Shokri work’s Equation (101), will **overlook the modification of the calibrated noise in omitted iterations**. Therefore, in proving Theorem 4.5, we use the coefficient $\Phi(k_1,k_2)$ to indicate the exact privacy loss decay rate.  Theoretically, based on Corollary 4.4, the privacy loss decreases exponentially if $i_0 \notin B_k^j$. By contrast, the approximation in Ye and Shokri’s work would have a looser RDP bound.
> > We also provide the **numerical comparison in Appendix D (line 1103)**  to compare their RDP bound and ours. The result shows our work provides a significantly tighter RDP bound than theirs.
> >
> > **5. Regarding small remarks on Lemma 4.1**
> >
> > Thanks for our remakers, the Lemma 4.1 is very simple and is widely used, we leave it for the continuity of our proof.
> >
> > **As for questions:**
> >
> > **Q1: About exponent $-1/L_T$:**
> >
> > The exponent of $-1/L_T^2$ and the proof of corollary 4.4 is explained in the response above.
> >
> > **Q2: Comparison with RDP bounds in Ye & Shokri and experiments on simple strongly convex models.:**
> >
> > As for the comparison between our bound and Ye’s RDP bound, we provide theoretical explanation and numerical comparison in answering the fourth point of weakness above.
> >
> > As illustrated in our response to the third point of weakness above, $\lambda$ in Ye & Shokri and $L_T$ in our framework have different motivations.
> >
> > As for logistic regression with L2-regularization, our algorithm will degrade to DP-SGD when $D=0$ and loss function is cross entropy. Hence, the numerical comparison provided in answering weakness 4 can be directly applied.The result shows our work provides a tighter RDP bound than theirs.
> >
> > [1] Ye J, Shokri R. Differentially private learning needs hidden state (or much faster convergence)[J]. Advances in Neural Information Processing Systems, 2022, 35: 703-715.

---

> > > ### Comment · Reviewer_Hu7P · 2024-11-25
> > >
> > > Thank you for the replies, especially for clarifying the relationship between $L_T$ and $\lambda$.
> > >
> > > I am still not really convinced. I suspect there is some error in the analysis, due to the following reasons:
> > >
> > > - In the new figure you added to the Appendix D (Fig. 4), the privacy costs are the same after one epoch, both for $\sigma=0.1$ and $\sigma=0.01$. The RDP of the Gaussian mechanism would be 10 times bigger in the former case. And there can't be much "privacy amplification by iteration" yet at that point.
> > >
> > > - Even if the results of that Fig. 4 are correct, the differences against the bound of Ye and Shokri are very small for $\sigma=0.1$ if you look at the y-axis.
> > >
> > > - I still don't understand why DP-SGD would give 10\% test accuracy on MNIST with appox. $(0.1,10^{-5})$-DP (I have converted that RDP to approx. DP) using MLP and approx. 50\% using LeNet. I'm quite sure it would be possible to get better results for that privacy budget.
> > >
> > > - Why would your method give so dramatically higher test accuracies for the same budget both for MLP and LeNet? The privacy costs cannot be much lower than those of DP-SGD (otherwise there is something wrong I think) and the layer-alternating optimization method cannot be that much better I think.
> > >
> > > Generally, for a given batch size, it is generally very difficult to get better privacy-utility for the final model using hidden state analysis than by using DP-SGD and composition analysis, this has also been shown recently in [Annamalai (2024)](https://arxiv.org/pdf/2407.06496). Thus I don't really think you can get that big differences, also considering that the layer-wise optimization method is likely generally worse than DP-SGD (or at least it cannot be that much better I think).
> > >
> > > I think the method is really interesting and has some value, but I just think the paper needs some more work. And I believe some of the results are not correct. Thus I will keep my score.
> > >
> > > Minor remark: the letter $D$ is used both for dataset and the number of layers.

---

> > > > ### Author Response · Authors · 2024-11-27
> > > > **Response to Question 1,2,3**
> > > >
> > > > Thanks for your further question. I hope our previous response answered your question in proofing techniques. We would like to further address your new questions in experiments one by one:
> > > >
> > > > **1. Quetion about Figure 4**
> > > >
> > > > Firstly, we acknowledge that we have mistakenly labeled the different lines in our chart's legend and the result is fixed in the updated version, but it does not affect our result. As for the result in the first epoch: The privacy loss in the first epoch would not be “ten times bigger” because **the variance of calibrated noise is not proportional to the privacy loss due to the decreasing term in the Langevin dynamic analysis framework** . More specifically, the privacy loss in every epoch is the average privacy loss of $\varepsilon_K(\alpha,j_0)$ in Equation (10), which is determined not only by the first term $\frac{\eta S_g^2}{4b^2 o(\eta, k, j_0)}$ but also by the decaying factor in the remaining part of the right hand side of Equation (10). The decay factor is calculated by the LSI constant during training, which also depends on the variance of the calibrated noise (Lemma A.3). Moreover, the new Figure also has Ye and Shokri’s RDP bound under different calibrated noise variance, and the first epoch does not have “ten times bigger” as well.
> > > >
> > > > **2. Question about the gap of RDP bound**
> > > >
> > > > The RDP gap depends on many parameters. In the new Figure,As $\sigma^2$ varies from 0.1 to 0.01, the RDP gap between two different kinds of accounting methods **becomes larger** . Moreover, if we adjust other parameters such as $\eta$ when $\sigma^2=0.001$, the gap would become **more obvious** , we also draw a [figure](https://anonymous.4open.science/r/iclr6700-666F/RDP.pdf) to prove our finding.
> > > >
> > > > Furthermore, we realize that carefully comparing the RDP bound between ours and Ye and Shokri’s work needs to analyze the dynamics of privacy loss w.r.t different hyperparameters, such as the learning rate, number of batches, which is not the main topic of this paper. In this paper, we mainly aim to derive the privacy loss bound for neural networks under the hidden state assumption. But we really thank the reviewer’s suggestion. As in the response to Question 4a from reviewer S7Yp, the dynamics of privacy loss w.r.t different hyperparameters may contribute to better adaptive calibrated noise strategy designing, which would be further explored in our future work.
> > > >
> > > > **3. Question about experiments in DP-SGD**
> > > >
> > > > Thanks for your question. We acknowledge that our previous experiments in DP-SGD did not fine-tuning the learning rate on various architectures and datasets. We have rerun the experiments about the DP-SGD part for the case you mentioned with extensively fine-tuned hyper-parameters in the updated version. The utility for DP-SGD dramatically improved. For example, the MNIST now has 80.09% average test acc. Despite improvement, the result of this case is consistent with previous observation, that is our algorithm (test acc is 94.36%) outperforms the DP-SGD. We have updated Appendix B.1 to include more detailed information.
> > > >
> > > > Furthermore, we have to emphasize that we employ the automatic clipping technique[2], which is normalized DP-SGD, as the benchmark for DP-SGD. Empirical evidence from [2] suggests that normalized DP-SGD outperforms or matches the state-of-the-art results without additional parameters such as clipping threshold. However, given that learning rate plays a big role in obtaining a good privacy loss in this context, it is tricky to pick a wise learning rate.
> > > >
> > > > Moreover, **for vanilla DP-SGD**[3], the trade-off between privacy and utility is more intricate. The privacy loss equals $(\alpha, K\times 2 S_g^2 \alpha/ \sigma^2) $-RDP for i.i.d Poisson sampling DP-SGD after $K$ iterations where the clipping threshold $S_g$ is crucial to the performance of DP models. Figure 1 in [2] shows that the utility varies a lot for different $S_g$, which cannot be derived from the privacy budget. Therefore, it would be necessary but computationally expensive to conduct 2D grid search for the learning rate and the clipping threshold jointly on vanilla DP-SGD.  In contrast, our algorithm can provide a converged privacy loss for various hyper-parameters, thereby easing the need for 2D hyper-parameter search.

---

> > > > > ### Author Response · Authors · 2024-11-27
> > > > > **Response to Question 4,5**
> > > > >
> > > > > **4. Question about the utility**
> > > > >
> > > > > Thanks for your question.  As shown in our paper Figure 1(Left) and Ye and Shokri’s paper Figure 1, the privacy loss under the hidden state assumption (HSA) is dramatically smaller than the composition theorem when the number of iterations $K$ is large. Conversely, under an equivalent privacy loss, the HSA requires much smaller calibrated noise. Given that calibrated noise is one of the key factors for utility degradation, the utility under the HSA can dramatically increase. It is also the key motivation why researchers[4] recently tried to introduce the privacy loss accountant under the HSA to DP-SGD algorithms.
> > > > > Instead of directly deriving the privacy loss accountant methods for DP-SGD under the HSA, our paper aims to achieve desirable properties under the HSA by modifying the algorithm. Therefore, we use DP-SBCD to train neural networks and derive privacy accountant methods, especially under the HSA. Although coordinate descent algorithms have utility loss in training neural networks compared with SGD, the algorithm may have **better utility when considering privacy**. Additionally, we believe that the coordinate descent approach may inherently possess privacy benefits. Intuitively speaking, the gradient of our algorithm is not w.r.t all parameters, which might mitigate the privacy loss associated with gradient leakage.
> > > > >
> > > > > **5. Question about hidden state assumptions and related paper**
> > > > >
> > > > > For comparison with paper [5]. [5] claims that a hidden state privacy amplification result for **DP-SGD** for all loss functions (especially non-convex functions mentioned in [5]) in general is not possible. But our paper’s algorithm is **DP-SBCD**, which decomposes the original problems as several convex problems. As we demonstrated in Problem 4, we think the coordinate descent type of method may offer distinct privacy advantages because DP-SBCD algorithm does not calculate the gradient w.r.t all parameters in one iteration. Thus, the disparity in utility between SGD and coordinate descent **does not convincingly predict** the utility difference between DP-SGD and DP-SBCD **under the same privacy loss budget**, especially given their different privacy accounting methods.
> > > > >
> > > > > Moreover, paper [5] purposely constructs a loss function that could preserve the likelihood ratio result in each iteration. In this way, the privacy amplification under the hidden state assumption does not improve the privacy much because the model in the last iteration preserves the same likelihood ratio as the first iteration.  However, as the author claimed in their conclusion, **It is currently unclear whether their techniques apply to commonly used functions such as ReLU activation functions.** That is to say, [5] considers the worst cases for hidden state assumption, which is different from our settings (like in terms of algorithm, activation function), it does not indicate hidden state assumption cannot improve privacy loss in the specific settings considered in our work.
> > > > >
> > > > >
> > > > > [1] Ye J, Shokri R. Differentially private learning needs hidden state (or much faster convergence)[J]. Advances in Neural Information Processing Systems, 2022, 35: 703-715.
> > > > >
> > > > > [2] Bu Z, Wang Y X, Zha S, et al. Automatic clipping: Differentially private deep learning made easier and stronger[J]. Advances in Neural Information Processing Systems, 2024, 36.
> > > > >
> > > > > [3]  Song S, Chaudhuri K, Sarwate A D. Stochastic gradient descent with differentially private updates[C]//2013 IEEE global conference on signal and information processing. IEEE, 2013: 245-248
> > > > >
> > > > > [4]Kong W, Ribero M. Privacy of the last iterate in cyclically-sampled DP-SGD on nonconvex composite losses[J]. arXiv preprint arXiv:2407.05237, 2024.
> > > > >
> > > > > [5]  It’s Our Loss: No Privacy Amplification for Hidden State DP-SGD With Non-Convex Loss. Meenatchi Sundaram Muthu Selva Annamalai, AISec 2024.

---

> > > > > > ### Author Response · Authors · 2024-12-02
> > > > > > **Looking Forward to the Reviewer's Reply.**
> > > > > >
> > > > > > Dear reviewer Hu7P,
> > > > > >
> > > > > > We thank again for your detailed comments and constructive suggestions. As the deadline approaches, we kindly ask if our responses have adequately answered your questions to address your concerns. We would greatly appreciate your feedback and discussion with you to ensure that we have fully resolved the outstanding issues for the improvement of this manuscript. Thank you for your time, effort and consideration.

---

> ### Comment · Reviewer_Hu7P · 2024-12-02
>
> Thank you for the replies! You wrote:
>
> > Thanks for your question. As shown in our paper Figure 1(Left) and Ye and Shokri’s paper Figure 1, the privacy loss under the hidden state assumption (HSA) is dramatically smaller than the composition theorem when the number of iterations $K$ is large. Conversely, under an equivalent privacy loss, the HSA requires much smaller calibrated noise. Given that calibrated noise is one of the key factors for utility degradation, the utility under the HSA can dramatically increase.
>
> I agree the DP guarantees can get much smaller, but in Ye and Shokri’s analysis this requires such a strong L2-regularization, that  it is very difficult to beat the privacy-utility of DP-SGD. Recently, Bok et al. (Bok, Jinho, Weijie J. Su, and Jason Altschuler. "Shifted Interpolation for Differential Privacy." ICML 2024) gave bounds in the hidden state analysis that are lower than those of Ye and Shokri. And their experiments on logistic regression already show that it is very difficult to beat the privacy-utility of DP-SGD
>
> You also write:
>
> > Furthermore, we realize that carefully comparing the RDP bound between ours and Ye and Shokri’s work needs to analyze the dynamics of privacy loss w.r.t different hyperparameters, such as the learning rate, number of batches, which is not the main topic of this paper. In this paper, we mainly aim to derive the privacy loss bound for neural networks under the hidden state assumption
>
> If I understand correctly from your replies, you haven't really carried out hyperparameter tuning for various methods. This might explain why DP-SGD's experimental results are so bad.
>
> I still think the idea is interesting, but also I still don't see why the method would be better than DP-SGD. I do not yet see convincing evidence for that, especially due to the fact that there are no experiments showing this.
>
> Also, if you look at the y-axis in Appendix Fig. 4, the differences are very small. The L2-regularization constant needed for Cor. 5.3 of Ye and Shokri seems to be missing.

---

> > ### Author Response · Authors · 2024-12-03
> > **Response to Reviewer Hu7P's Question 1**
> >
> > Thanks for your reply, we would like to response to your questions one by one:
> >
> > **Q1: The privacy-utility trade-off between DP-SGD under the composition theorem and DP-SBCD under the hidden state assumption(HSA)**
> >
> > Thanks for your question. To clarify, we **do not** claim DP-SBCD can **always** achieve a better privacy-utility trade-off than DP-SGD and its variants **under any circumstance**. Actually, the focus of our paper is neural network training, where the loss functions are **non-convex**. By contrast, the paper you mentioned [1] investigates privacy loss under the hidden state assumption (HSA) for **convex loss functions**. Moreover, the advantages of hidden state assumption (HSA), compared with the composition theorem, are more obvious with a larger number of training iterations, which is usually necessary for solving more complex non-convex problems.
> >
> >
> > To further elaborate, the theoretical results have indicated that DP-SGD’s convergence rate for convex loss functions and non-convex loss functions are different, as indicated in the table below where $n$ is the number of samples, $d$ is the dimension of the model, and $\varepsilon$ is the privacy loss.
> >
> > | condition for loss function                    | bound |
> > |------------------------------|-------|
> > | convex                       | $O(\sqrt{\frac{1}{n}}+\frac{\sqrt{d}}{\varepsilon n})$  [3] |
> > | ($L_0,L_1$)-Gereralized Smooth | $O(\sqrt[4]{\frac{d}{n^2\varepsilon^2}})$ [4]  |
> >
> >
> > For the cases with a convex loss function, such as logistic regression, it is possible that the DP-SGD under composition theorem may provide a better privacy-utility trade-off compared with DP-SBCD under the hidden state assumption because the training converges very fast and the number of iterations is small. This is also indicated in Ye and Shokri’s work [2].
> >
> > However, in the case of deep neural network training, DP-SGD with non-convex loss function usually **has a slower convergence rate**, making it necessary to have a large number of iterations to achieve good utility. Therefore, the privacy loss accounting methods under the hidden state assumption (HSA) **offer more advantages** over the composition theorem in terms of the utility-privacy trade-off compared with convex problems. Although we believe our proposed DP-SBCD may not be the **final solution** to derive a tight privacy loss guarantee for non-convex problems under the HSA, there are **no privacy loss accountant methods for training nonlinear neural networks under the HSA** currently. Exploring and improving methods in this context need more effort and we include them in the future works.
> >
> >
> >
> > **In summary, the focus and the main contribution of this work is to derive a tight privacy loss guarantee for neural network training under the hidden state assumption (HSA)**. Previously, the privacy loss accountant methods designed specifically for the HSA, such as [1, 2], are applicable only for convex loss functions. We acknowledge there may be some room for improvement in terms of privacy loss under the HSA for non-convex loss objective functions, which is an interesting question worth further exploration.

---

> > > ### Author Response · Authors · 2024-12-03
> > > **Response to Reviewer Hu7P's Question 2**
> > >
> > > **Q2: About hyper-parameter tuning**
> > >
> > > Our experiments did hyper-parameter tuning in the revised version, and we provided a detailed ablation study below
> > >
> > > | learning rate       | $1/2^0$ | $1/2^1$ | $1/2^2$ | $1/2^4$ | $1/2^7$ |
> > > |---------------------|-------|-------|-------|-------|-------|
> > > | MNIST (LeNet)       | 74.91 | 79.40 | 70.41 | 60.10 | 4.97  |
> > > | MNIST(MLP)          | 10.00 | 10.00 | 10.00 | 74.61 | 30.71 |
> > > | FasionMNIST (LeNet) | 74.89 | 71.84 | 70.69 | 64.26 | 6.54  |
> > > | FasionMNIST (MLP)   | 10.00 | 10.00 | 10.00 | 75.08 | 17.97 |
> > > ||
> > >
> > >
> > >
> > > Moreover, we have to emphasize that we use auto-clipping DP-SGD, which is a variant of vanilla and which **provides state-of-the-art results without tuning the clipping threshold**, so the results in the table above only exploit different learning rates. For comprehensiveness, we also provide an ablation study here for LeNet MNIST for vanilla DP-SGD where $C$ is the clipping threshold. More results on different datasets can also be found in [5]. Please be noted that it is time consuming and tricky to tune both the learning rate and the clipping threshold jointly in practice.
> > >
> > > |       | lr=$1/2^0$ | lr=$1/2^1$ | lr=$1/2^2$ | lr=$1/2^4$ | lr=$1/2^7$ |
> > > |-------|----------|----------|----------|----------|----------|
> > > | $C=0.01$ | 73.05    | 70.37    | 70.56    | 64.56    | 10.26   |
> > > | $C=0.1$ | 75.15    | 69.19    | 67.92    | 62.24    | 13.76   |
> > > | $C=1$   | 73.23    | 72.91    | 68.79    | 58.28    | 16.60    |
> > > | $C=10$  | 71.22    | 73.01    | 70.56    | 62.78    | 8.88     |
> > > | $C=100$ | 74.63    | 72.41    | 69.83    | 59.48    | 15.89  |
> > > ||
> > >
> > >
> > > Our previous response intends to emphasize that the RDP bound compared with Ye and Shokri’s work **is not the main contribution of this work** and further comparison requires more effort. Frankly speaking,  we mentioned the RDP gap between Ye’s work and ours only once in line 415, which is because we want to show that we have different proof techniques and our proof technique eliminates the approximation part they have. Our main contribution is that **we designed the DP-SBCD algorithm and privacy loss accountant method specifically for the hidden state assumption in training neural networks.**
> > >
> > > [1] Bok J, Su W, Altschuler J M. Shifted Interpolation for Differential Privacy[J]. arXiv preprint arXiv:2403.00278, 2024.
> > >
> > > [2] Ye J, Shokri R. Differentially private learning needs hidden state (or much faster convergence)[J]. Advances in Neural Information Processing Systems, 2022, 35: 703-715.
> > >
> > > [3] Bassily R, Feldman V, Talwar K, et al. Private stochastic convex optimization with optimal rates[J]. Advances in neural information processing systems, 2019, 32.
> > >
> > > [4] Yang X, Zhang H, Chen W, et al. Normalized/clipped sgd with perturbation for differentially private non-convex optimization[J]. arXiv preprint arXiv:2206.13033, 2022.
> > >
> > > [5] Bu Z, Wang Y X, Zha S, et al. Automatic clipping: Differentially private deep learning made easier and stronger[J]. Advances in Neural Information Processing Systems, 2024, 36.

---

> > > > ### Author Response · Authors · 2024-12-03
> > > > **Look Forward to Your Reply**
> > > >
> > > > Dear reviewer Hu7P,
> > > >
> > > > We thank again for your detailed comments and constructive suggestions. As there are only a few hours left for the author-reviewer discussion, we would like to kindly ask if your remaining concerns have been adequately addressed. Of particular note, we wish to emphasise that the primary contribution of our work lies in our pioneering effort to extend privacy loss accountant method for hidden state assumption and non-convex loss functions,  thereby encompassing a significantly broader range of practical scenarios than previous work studying (strongly) convex loss functions. This major contribution is highlighted in our manuscript as well as the responses to many reviewers.
> > > >
> > > > We would greatly appreciate your feedback and thank you for your time, effort and consideration.

---

> > > > > ### Comment · Reviewer_Hu7P · 2024-12-03
> > > > >
> > > > > Thank you for the replies. It is quite hard to judge based on these numerical results. It looks to me that even larger learning rate could have given better results. I agree the method is very inteteresting, and I believe the analysis is solid, though it very strongly relies on the previous analysis by Ye and Shokri (some of the proofs are almost copies of the proofs by Ye and Shokri).
> > > > >
> > > > > I think it is unfortunate that there are no clear experimental comparisons against DP-SGD, to be honest it is a bit difficult to judge how comprehensive / accurate these comparisons are. I still think that DP-SGD should give better results for the stated privacy budgets.
> > > > >
> > > > > Due to the novelty of the method, I am raising my score, but I am still leaning towards a reject.

---

> > > > > > ### Author Response · Authors · 2024-12-04
> > > > > > **Response to Reviewer Hu7P about the baseline**
> > > > > >
> > > > > > Thanks for your response and acknowledgment!
> > > > > >
> > > > > > We realize that your concern still focuses on the baseline methods. We find a paper[1] that provides a baseline **vanilla DP-SGD** where the privacy budget is $\varepsilon=0.3$, which is almost twice as much as ours, and their test accuracy is 93.29%[1], which is still smaller than our algorithm.
> > > > > >
> > > > > > Their experiment result also reminds us that there is a trade-off between utility and the number of epochs, and we also search for the best number of epochs for LeNet MNIST, the primary result is as follows:
> > > > > >
> > > > > > | number of epochs   | 50 | 100 | 200 | 300 |
> > > > > > |---------------------|---------|---------|---------|------|
> > > > > > | DP-SBCD      | 95.85 | 96.34 | 97.01 |97.17|
> > > > > > | DP-SGD        | 74.91 | 91.08 | 94.40| 94.76|
> > > > > >
> > > > > > The result indicates that with more epochs, the gap between the DP-SGD and DP-SBCD becomes smaller, **but the result still holds** as our algorithm outperforms the DP-SGD. Further experiments will be updated in the revised version because more epochs require more training time.
> > > > > >
> > > > > > Thanks again for your question and efforts!
> > > > > >
> > > > > > [1] Feldman V, Zrnic T. Individual privacy accounting via a renyi filter[J]. Advances in Neural Information Processing Systems, 2021, 34: 28080-28091.

---

### Official Review · Reviewer_dtdh · 2024-11-03

**Soundness:** 3
**Presentation:** 3
**Contribution:** 2
**Rating:** 6
**Confidence:** 3

**Summary:**

The paper proposes a privacy mechanism for non-convex models by transforming it into a sub-convex one, offering privacy guarantees through this relaxation.

**Strengths:**

The approach is well-organized and is presented in a clear manner.

**Weaknesses:**

I have some concerns regarding the motivation and advantages of this method compared to the classical DP-SGD algorithm.

DP-SGD is applicable to non-convex problems and does provide privacy guarantees at stable points. How does the proposed method theoretically and empirically compare to DP-SGD in terms of privacy and utility? The relaxation from a non-convex to a convex problem in the proposed method could impact utility. How does the utility of the proposed method compare to DP-SGD, especially in terms of balancing privacy and performance?  Whether the relaxation to a sub-convex problem offers distinct privacy advantages or if it primarily facilitates privacy analysis? Empirical evidence highlighting this trade-off could help demonstrate the practical benefits or limitations of this approach.

I may adjust my evaluation score depending on these responses.

**Questions:**

1. There are several typos. E.g., ", which" on line 174.
2. Line 193: use text descriptions instead of math symbols directly in the text wherever applicable. E.g., "for all d"

---

> ### Author Response · Authors · 2024-11-20
> **Response for weakness and Questions**
>
> Thanks for your question, As for the weakness part, we would like to answer your questions one by one:
>
>
> **1. How to compare with classical DP-SGD in terms of privacy and utility?**
>
> Our paper’s privacy loss analysis considers the hidden state assumption, which assumes that the intermediate states are hidden to adversaries. Previous work [1,2,3] shows that privacy loss under this assumption has some benign results such as smaller and converged privacy loss. However, these nice properties cannot be applied to neural network training using existing works, because these works require a strongly convex loss function. In order to obtain a tighter privacy loss estimation under the hidden state assumption for neural network training, we propose DP-SBCD algorithm and the corresponding privacy loss accountant in this paper.
>
> Since there is no privacy loss accountant method for DP-SGD designed specifically for the hidden state assumption, when comparing the privacy and utility with our method, we have to use composition theorem to derive the privacy loss for DP-SGD. In this context, we provide the utility under the same privacy loss in Table 2 of Appendix B.2. Intuitively speaking, based on Corollary 4.4, the privacy loss for our algorithm under the hidden state assumption increases only once for each epoch and decreases for the rest iterations. **The tight privacy loss accountant confers an advantage for DP-SBCD over DP-SGD in terms of utility with the same privacy loss.**
>
> On the other hand, as we discussed in line 468 to 476, directly comparing the privacy loss under two different assumptions and two different types of algorithms is **unfair** since these two kinds of algorithms share different hyper-parameters such as learning rate, batch size, which directly contribute to the privacy loss. In this regard, it is necessary but challenging to derive a tight privacy loss estimation method for DP-SGD under the hidden state assumption, which is, however, out of this paper’s scope and left as future works. The contribution of our paper is that **we first solve an important but challenging problem, i.e., the privacy accountant under the hidden state assumption, in the context of deep neural networks**.
>
> **2. The relaxation from a non-convex to a convex problem in the proposed method could impact utility. How does the utility of the proposed method compare to DP-SGD, especially in terms of balancing privacy and performance**
>
> The relaxation from a non-convex to a convex problem does not necessarily impact the utility. The reason why our algorithm could improve the utility compared with the DP-SGD is because our algorithm applies the privacy loss accountant under the hidden state assumption. As we explained above, our algorithm requires smaller calibrated noise compared with DP-SGD whose privacy loss is calculated by composition theorem under the same privacy budget. That’s the reason why our experiments in Table 2 of Appendix B.1 show that our algorithm provides better utility compared to DP-SGD.
>
>
> **3. Whether the relaxation to a sub-convex problem offers distinct privacy advantages or if it primarily facilitates privacy analysis?**
>
> We are not sure whether the relaxation to sub-problem offers distinct privacy advantages or not because it is out of the scope of this work. The DP guarantees now are always based on privacy analysis rather than employing the DP guaranteed model to different attack methods such as membership inference attack. Generally speaking, the failure rate of membership inference attacks and differential privacy guarantees indicate the upper bound and the lower bound of privacy, respectively. The **exact privacy** of a mechanism is usually intractable due to its complexity, and our work aims to derive a tight lower bound of privacy. Intuitively speaking, we think the coordinate descent type of method offers distinct privacy advantages because we do not calculate the gradient w.r.t all parameters. Moreover, our algorithm does facilitate privacy analysis because each subproblem is convex, which provides a guaranteed bound of the Lipschitz constant $L_F$ of the update scheme function.
>
> As for questions, thanks for your comment, the typos mentioned have been fixed in the updated version.
>
> [1] Ye J, Shokri R. Differentially private learning needs hidden state (or much faster convergence)[J]. Advances in Neural Information Processing Systems, 2022, 35: 703-715.
>
> [2]Feldman V, Mironov I, Talwar K, et al. Privacy amplification by iteration[C]//2018 IEEE 59th Annual Symposium on Foundations of Computer Science (FOCS). IEEE, 2018: 521-532.
>
> [3]Chourasia R, Ye J, Shokri R. Differential privacy dynamics of langevin diffusion and noisy gradient descent[J]. Advances in Neural Information Processing Systems, 2021, 34: 14771-14781.

---

> ### Comment · Reviewer_dtdh · 2024-11-23
>
> Thank for your response. The main contribution appears to be the application of block gradient descent to transform the original problem into a sub-optimal convex problem, enabling the use of standard privacy analysis techniques. While the idea is new, it does not strike me as particularly compelling. Shokri's work, which employs Langevin diffusion to trace the distribution of perturbed data, could potentially be applied to the sub-optimal convex problem as well. I will raise my score since the idea is new, though I am not entirely confident in my assessment. Overall, the approach does not feel particularly innovative to me.

---

> > ### Author Response · Authors · 2024-11-23
> > **Further clarification about our contribution**
> >
> > Thanks for your acknowledgement!  We would like to further highlight our work’s contribution in algorithms and privacy loss analysis.
> >
> > **1. [Regarding sub-optimal problems and framework generality]**:  As for suboptimal problems: It is true that Ye&Shokri’s work could be potentially applied to sub-optimal problems. However, as demonstrated in their Corollary 5.3  it is also **necessary** for them to have a regularization term to ensure that the overall loss objective is strongly convex. By contrast, the regularization term in our framework is **optional**. When there is no regularization, we can simply let $L_T = 1$ and our method still works. In addition, our framework supports *any regularization term, as long as it is convex* (in contrast to *strongly-convex ones* required in Ye & Shokri), which means our framework includes more regularization schemes such as lasso. Fundamentally, we can obtain feasible privacy loss under the hidden state assumption because the update rules mapping the old parameters to the new parameters (e.g. function $T$ and $F$ in Equation (9)) have bounded Lipschitz constants. **Such bounded Lipschitz constants are achieved in different ways in Ye & Shokri and our framework**. In Ye & Shokri, it is achieved by introducing an $l_2$ regularization scheme to make the loss function of regularized logistic regression strongly convex (See Equation (7) in Ye & Shokri). By contrast, we decompose deep neural network training into several sub-problems and each of these sub-problems have a convex loss function, this guarantees the bounded Lipschitz constant for the update rule (See Lemma 4.1 of our work which bounds $L_F$). The regularization scheme is optional and more generic in our framework. We discuss the regularization terms in line 304 to 308 to demonstrate our framework is generally applicable. In a nutshell, **both Ye & Shokri and our work manage to bound the Lipschitz constant of the update rule so that we can have a tight privacy loss estimation under the hidden state assumption, but the techniques used in our work to achieve Lipschitz bound are totally different from Ye & Shokri**
> >
> > **2. [Regarding the privacy loss analysis and the tightness of the bound]**: We would also like to highlight our contribution to privacy loss analysis. Firstly, our algorithm provides the privacy loss accountant method for **adaptive calibrated noise** under the hidden state assumption. The calibrated noise setting is **more general**, it explains why privacy loss could converge under the hidden state assumption, as we demonstrated in Section 4.3. Moreover, our experimental results also show that adaptive calibrated noise has the potential to provide a better privacy-utility trade-off.
> >
> > Moreover, employing adaptive calibrated noise motivates us to derive **a tighter RDP bound**. When the calibrated noise is adaptive, we need to carefully track the privacy loss among each iteration, since the approximation error would be exaggerated in iterations with a lower-magnitude calibrated noise. In our proof, we  use the coefficient $\Phi(k_1,k_2)$ (Equation (27) in our work) to indicate the exact privacy loss decay rate.  By contrast, Ye and Shokri work’s using part of iterations' privacy loss to approximate the whole epoch’s privacy loss (their work’s Equation (101)), will overlook the omitted iterations’ contribution in total privacy loss. Therefore, our RDP bound is tighter.  We provide the numerical comparison in Appendix D(line 1103) to compare their RDP bound and ours. The result shows our work provides a significantly tighter RDP bound than theirs.
> >
> >
> > We hope our further explanation could address your concern. And thanks again for your comment!

---

### Official Review · Reviewer_bmCR · 2024-11-04

**Soundness:** 4
**Presentation:** 2
**Contribution:** 3
**Rating:** 6
**Confidence:** 2

**Summary:**

This paper proposes DP-SBCD, which train neural networks with theoretical guarantees of DP under hidden state assumption

**Strengths:**

The paper theoretically proved the connection between DP-SBCD and its DP property.

**Weaknesses:**

Over-sale for the contribution. DP-SGD is feasible in many cases. For example, the number of learning steps is not extremely large.

**Questions:**

What is the complexity of the proposed algorithm?

---

> ### Author Response · Authors · 2024-11-20
> **Response for weakness and questions**
>
> Thanks for your comments and questions!  Below are our point-to-point responses:
>
> **As for the weakness:**
>
> Typically, neural network training requires many iterations to obtain a well-trained model. Considering the common practice of only saving the final model checkpoints, the privacy loss derived by the composition theorem **will be significantly overestimated** when we have many iterations during training. We agree that DP-SGD is more generally applicable, but there is no tight privacy loss estimation method for DP-SGD and non-convex loss functions specifically designed for the hidden state assumption so far. That is to say, for DP-SGD and non-convex loss functions, we have to use the composition theorem to obtain the **over-pessimistic privacy loss**, which becomes infeasible for large numbers of iterations. To derive a tight privacy loss estimation for DP-SGD and non-convex loss function under the hidden state assumption will be an important future work.`
>
>
> **As for the complexity:**
>
>  **For time complexity:** For one epoch, coordinate descent algorithms share the same time complexity as gradient descent. However, the DP-SGD requires per-sample gradient clipping, which requires the **use of back-propagation for each sample**. Our algorithm only clips the auxiliary variable $x_d$. Hence, our algorithm outperforms the DP-SGD in time complexity, especially under a large batch size.
>
> **For memory complexity:** Coordinate descent is designed to provide a memory-friend algorithm in solving large scale problems. For each iteration, our algorithm only updates one layer with **one-layer back-propagation**. But the DP-SGD requires updating all parameters with the **whole-layer back-propagation**. Hence, our algorithm is more memory efficient as well.

---

> > ### Author Response · Authors · 2024-11-30
> > **Follow-up after response**
> >
> > Thanks again for your time and comments. We just want to reach out to see if our response addresses your concerns. We are also happy to discuss any further questions or comments that you may have after our response.

---

### Official Review · Reviewer_S7Yp · 2024-11-07

**Soundness:** 3
**Presentation:** 2
**Contribution:** 3
**Rating:** 6
**Confidence:** 2

**Summary:**

This paper introduces the differentially private stochastic block coordinate descent (DP-SBCD) algorithm for training neural networks with differential privacy guarantees under the hidden state assumption. This method decomposes the training process into sub-problems for each network layer, allowing for a more precise privacy analysis. By utilizing adaptive noise distributions, DP-SBCD achieves a better trade-off between model utility and privacy. The theoretical analysis demonstrates that privacy loss converges under the hidden state assumption, making it suitable for non-convex and non-smooth loss functions in neural networks. Empirical results show that DP-SBCD has lower privacy loss and better utility compared to traditional differentially private stochastic gradient descent (DP-SGD) methods.

**Strengths:**

- The authors propose differentially private stochastic block coordinate descent (DP-SBCD), which solves non-convex problems with differential privacy guarantees under the hidden state assumption.
- The authors provide insights to understand the convergence of the privacy loss in Theorem 4.5 and Section 4.3.
- The limitations are well discussed. The authors explain that although DP-SBCD shows lower privacy loss than DP-SGD, direct comparisons can be misleading due to different privacy loss assumptions. They also discuss the memory efficiency of the coordinate descent algorithm and the challenges it introduces, such as high variance in mini-batch gradients and the need for large batch sizes to mitigate this.

**Weaknesses:**

Please refer to the questions.

**Questions:**

- For the log-Sobolev inequality (LSI) for the distribution of $\theta$ in Lines 309-310 and Theorem 4.5:

1. Is the LSI proven based on Assumption 4.3, or is it an additional assumption?
2. What does the constant $c$ in Line 350 represent, and how is it determined?
3. Given that previous works assumed strongly convex smooth loss functions, how do the authors justify the LSI assumption in this non-convex neural network problem?
4. How does the subproblem viewpoint contribute to making the LSI assumption reasonable in this context?

- What causes the fluctuations in the unseen privacy loss shown by the dashed green and red lines in Figure 1 (right)?

- For the sentence "Otherwise, the privacy loss will increase exponentially" in Lines 78-79?  Are the authors referring to DP-SGD or other methods in this statement? I.e., the DP-SGD leads to exponentially increasing privacy loss in settings that are not strongly convex or smooth?

- For Figure 2 (a):

1. Is there a theoretical explanation for why the curves nearly intersect at K=30?
2. How does this intersection point relate to the convergence of privacy loss discussed in the paper?
3. Can the authors provide a prediction or additional plot showing the behavior of these curves for K>30?
4. Are there any practical implications of this intersection point for choosing the number of epochs in real-world applications?

---

> ### Author Response · Authors · 2024-11-20
> **Response for Question 1,2,3**
>
> Thanks for your constructive comments, we answer your questions one by one:
>
>
> **1. For a series of problems related to the LSI, we would like to first review the previous papers and introduce why we requires LSI constant:**
>
> The mathematical definition of the LSI is demonstrated in Definition A.1 of Appendix A.4 (line 697 - 701). The LSI assumption is related to the distribution of the parameter space and is independent of the loss function. The LSI is a very mild assumption since strongly logconcave distributions, such as Gaussian distribution and uniform distribution, satisfy the LSI assumption. Moreover, [1] indicates that the LSI can also be satisfied by some non-logconcave distributions.Then, we would like to answer the sub-questions one by one:
>
>
> **a)** Yes, the LSI is an additional assumption.
>
> **b)** The constant in Line 350 is the LSI constant $c$ with the detailed definition in Equation (13) in Definition A.1 of Appendix A.4 (line 697 - 701). Based on Equation (4) in [1], the LSI constant of a distribution $\mu$ is related to the lower bound of the ratio of the KL divergence and relative Fisher information for any other distribution $\rho$. For strongly logconcave distribution $\mu$, if $- \log \mu$ is $c$-strongly convex, then $\mu$ satisfies LSI with constant $c$. For example, the tightest LSI constant for standard Gaussian distribution is $1$. In our analysis, the LSI constant of the parameter distribution is calculated in Lemma A.3 in Appendix A.5, it helps us to bound the gradient of the Renyi divergence and thus the privacy loss. Specially, we initialize the LSI constant as $0$ in the beginning in line 854. It is because we consider the $\nu$, which is the distribution for the adjacent dataset $D’$, as the other distribution in the definition of LSI. Therefore, in the initialization point, the two distributions are the same. Hence, the $c_0^0=\infty$ based on Equation (4) in [1]. The same initialization has also been used by [2].
>
> **c)** The strongly convexity assumption in previous work [2] means the loss objective is a strongly convex function w.r.t. parameters. while the LSI assumes the distribution of parameters, which are totally different and independent. As mentioned above, the LSI is a mild condition on parameters; it can be easily satisfied if we initialize the parameter with Gaussian distribution or uniform distribution, which are the most popular initialization schemes.
>
> **d)** As we mentioned above, the LSI is a very mild assumption on the distribution of parameter space. Moreover, the parameter is initialized i.i.d. Hence, the subproblems also satisfy the LSI assumption.
> We realize that our paper should provide more details about the LSI assumption; we have improved this part in the revised version (from line 312 to line 315) .
>
> **2. What causes the fluctuations in the unseen privacy loss shown by the dashed green and red lines in Figure 1 (right)?**
>
> Corollary 4.4 shows that **for each epoch**, the privacy loss will increase once when $i_0 \in B_k^j$ and decrease otherwise. Figure 1 graphically represents this dynamic behavior. The distinct colors employed in the figure correspond to different values of $i_0$ as delineated in Corollary 4.4. Moreover, as indicated in line 934 to 942, the final privacy loss would be the average over different cases of $i_0$, which will be smoother than the red/green dash lines in Figure1(right).
>
> **3. For the sentence "Otherwise, the privacy loss will increase exponentially" in Lines 78-79? Are the authors referring to DP-SGD or other methods in this statement? I.e., the DP-SGD leads to exponentially increasing privacy loss in settings that are not strongly convex or smooth?**
>
> Sorry for the confusing statement. In Lines 78-79, we intend to point out that the privacy loss of DP-SGD under the hidden state assumption will exponentially increase if we follow the same analysis framework as in [2]. It does not mean that the actual privacy loss of the DP-SGD would increase exponentially. I have clarified this in our revised manuscript (line 79).
>
> When the loss function is non-convex, the Lipschitz constant of the update scheme (i.e., $L_F$ in our analysis) will be larger than 1. The privacy loss derived in the framework of [2] will thus increase exponentially with the number of update iterations.
>
> [1] Vempala S, Wibisono A. Rapid convergence of the unadjusted langevin algorithm: Isoperimetry suffices[J]. Advances in neural information processing systems, 2019, 32.
>
> [2] Ye J, Shokri R. Differentially private learning needs hidden state (or much faster convergence)[J]. Advances in Neural Information Processing Systems, 2022, 35: 703-715.

---

> > ### Author Response · Authors · 2024-11-20
> > **Response for Question 4**
> >
> > **4. For Problems in Figure 2(a):**
> >
> > Since Corollary 4.4 shows that **for each epoch**, the privacy loss will increase only once when $i_0 \in B_k^j$ and decrease otherwise. We believe the reviewer may misunderstand the meaning of the y-axis of Figure 2 (a). Based on Equation (10) in Theorem 4.5, each epoch during training has its contribution to the overall privacy loss under the hidden state assumption, which corresponds to each component on the right hand side of Equation (10) and depends on the number of total epochs. Curves in Figure 2 (a) represent the contribution of each epoch to the overall privacy loss under **a fixed number** of the total epoch (which is 30 in this case). Unlike the composition theorem, the privacy loss contribution of each epoch in the hidden state assumption depends on the total number of epochs, because only the initial and the final states will be revealed to the adversary in this case. Therefore, the curves in Figure 2 (a) have no implications for the privacy loss convergence. In addition, we need to re-calculate all the points in the curves if the total number of epochs during training changes.
> > Based on the clarification above, we answer the reviewer’s sub-problems one by one below:
> >
> > **a)** First, the curves in Figure 2(a) are not intersected exactly at $k=30$, nevertheless there are theoretical explanations about the intersection near $k=30$. The points with x-axis value $k$ on the curves in Figure 2(a) are calculated based on the logarithm (since Figure 2(a) uses log-scale for y-axis) of the $k$-th component inside the summation symbol on the right hand side of Equation (10). We calculate the gradient of the logarithm of this component w.r.t $k$, the result shows that when $L_F$ close to 1, the gradient is positive and approximately proportional to $j_0$. It illuminates why curves with a larger $j_0$ have a larger slopes.  Meanwhile, in the early phase such as $k=0$, larger $j_0$ have smaller contributions to the total privacy loss. Larger slopes combined with smaller initial values cause the intersection of these curves.
> >
> > The dynamics of privacy loss for different $j_0$ is not the main topic of this paper so this paper does not investigate how different hyper-parameters affect the privacy loss dynamics. Figure 2(a) intends to **show that the last few epochs contribute to the overall privacy loss**. But we really thank the reviewer's suggestion for considering the privacy loss trajectory among different $j_0$, which may contribute to better adaptive calibrated noise strategy designing, which would be further explored in our future work.
> >
> > **b)** We would like to clarify that these curves in Figure 2(a) do not strictly intersect at the last epoch. In addition, this observation is not related to the convergence of privacy loss discussed in the paper, because the curves in Figure 2(a) represent the contributions of each epoch to the privacy loss for **a fixed number** of total epochs (which is 30 in Figure 2(a)). On the other hand,the convergence of privacy loss studies how the privacy changes with **different numbers** of total epochs, where we need to calculate the privacy loss of different epoch numbers. (which is 1 to 30 in Figure2(b))
> >
> > **c)** We would like to clarify that curves in Figure 2(a) represent the contributions of each epoch to the privacy loss for **a fixed number of total epochs** under the hidden state assumption. For all cases in Figure 2(a), the number of epochs is fixed as 30. Unlike the composition theorem, the privacy loss contribution of each epoch under the hidden state assumption depends on the total number of epochs, since the adversary only has access to the initial and the last states of training. Therefore, if we change the total number of epochs and draw the behavior for $K>30$, we need to recalculate all data. Moreover, we believe the curves under these cases would be qualitatively the same as Figure 2(a).
> >
> > **d)** All curves on Figure 2(a) are using the same number of total epochs. Although we believe the curves will converge similarly as Figure 2(a) for different values of total number of epochs, i.e., $K$, it does not provide any guidance for real-world applications. The appropriate number of epochs in real-world applications depends on other factors as well, such as the magnitude of the calibration noise and the learning rate.

---

> > > ### Comment · Reviewer_S7Yp · 2024-11-26
> > >
> > > Thank you for your efforts and clarification. I have slightly raised my score.

---

> > > > ### Author Response · Authors · 2024-11-30
> > > >
> > > > Thanks for your acknowledge! We are also happy to discuss any further questions or comments if you may have!

---

### Author Response · Authors · 2024-11-20
**General response about the Hidden State Assumption**

We sincerely thank all reviewers for their efforts on this paper. But we realize that some of the reviewers miss one important assumption, we would like to provide some further explanation:

Our paper’s privacy loss analysis considers the hidden state assumption, which assumes that **the intermediate states are hidden to adversaries**. Previous work [1,2,3] shows that privacy loss under this assumption has some benign results. For example, [2] and Corollary 4.4 in our paper shows that privacy loss increases only once in each epoch and then starts to decrease, which aligns with our intuitive understanding. Therefore, algorithms under this privacy loss accountant require smaller calibrated noise, which on the other hand, increases algorithms’ utility.  Moreover, [1,2,3] indicates that privacy loss can converge under several epochs.

However, these benign results require a convex loss function in previous literature, which hindered its application in the deep learning community. In our paper, we further explored the privacy loss accountant methods under the hidden state assumption and proposed the DP-SBCD algorithm that **can enjoy the above properties in solving non-convex problems**. Moreover, in exploring the loss accountant method under the hidden state assumption, we also reveal the reason why privacy loss can converge under such assumption, and further improved the original privacy loss accountant methods. We attach a figure to show our improvement in privacy loss accountant compared with [1] in our updated version in the Appendix D.

[1] Ye J, Shokri R. Differentially private learning needs hidden state (or much faster convergence)[J]. Advances in Neural Information Processing Systems, 2022, 35: 703-715.

[2]Feldman V, Mironov I, Talwar K, et al. Privacy amplification by iteration[C]//2018 IEEE 59th Annual Symposium on Foundations of Computer Science (FOCS). IEEE, 2018: 521-532.

[3]Chourasia R, Ye J, Shokri R. Differential privacy dynamics of langevin diffusion and noisy gradient descent[J]. Advances in Neural Information Processing Systems, 2021, 34: 14771-14781.

---

### Meta-Review · Area_Chair_zCwX · 2024-12-22

**Metareview:**

The paper proposes a method for training multi-layer neural networks with differential privacy (DP) guarantees in the hidden-state model of DP. This is obtained using a convex dual formulation of the network (Zeng et al., 2019) and by using the privacy amplification by iteration analysis (Feldman et al., 2018).

This paper is on the borderline. All reviewers mention the paper provides a reasonable solution for training a model with non-convex objectives with DP. However, the last reviewer mentioned their proofs are almost copies of the proofs by Ye and Shokri (lowering the novelty of their proofs) and the DP-SGD comparison is not convincing. The authors tried to use the "faster convergence" argument to back up why their method is better than DP-SGD, while the reviewer claims that even larger learning rates could have given better results for the suggested method. If the authors address these issues properly, I believe it will make a great next submission.

**Additional Comments On Reviewer Discussion:**

the last reviewer mentioned their proofs are almost copies of the proofs by Ye and Shokri (lowering the novelty of their proofs) and the DP-SGD comparison is not convincing. The authors tried to use the "faster convergence" argument to back up why their method is better than DP-SGD, while the reviewer claims that even larger learning rates could have given better results for the suggested method.

---

### Decision · Program_Chairs · 2025-01-22

Reject